# Common variation in meiosis genes shapes human recombination and aneuploidy

Sara A. Carioscia[1,4], Arjun Biddanda[1,4], Margaret R. Starostik[1], Xiaona Tang[1], Eva R. Hoffmann[2], Zachary P. Demko[3] & Rajiv C. McCoy[1✉]

The leading cause of human pregnancy loss is aneuploidy, often tracing to errors in chromosome segregation during female meiosis[1,2]. Although abnormal crossover recombination is known to confer risk for aneuploidy[3,4], limited data have hindered understanding of the potential shared genetic basis of these key molecular phenotypes. To address this gap, we performed retrospective analysis of pre-implantation genetic testing data from 139,416 in vitro fertilized embryos from 22,850 sets of biological parents. By tracing transmission of haplotypes, we identified 3,809,412 crossovers, as well as 92,485 aneuploid chromosomes. Counts of crossovers were lower in aneuploid versus euploid embryos, consistent with their role in chromosome pairing and segregation. Our analyses further revealed that a common haplotype spanning the meiotic cohesin *SMC1B* is associated significantly with both crossover count and maternal meiotic aneuploidy, with evidence supporting a non-coding *cis*-regulatory mechanism. Transcriptome- and phenome-wide association tests also implicated variation in the synaptonemal complex component *C14orf39* and crossover-regulating ubiquitin ligases *CCNB1IP1* and *RNF212* in meiotic aneuploidy risk. More broadly, variants associated with aneuploidy often showed secondary associations with recombination, and several also exhibited associations with reproductive ageing traits. Our findings highlight the dual role of recombination in generating genetic diversity, while ensuring meiotic fidelity.

Despite their critical role in encoding genetic information, chromosomes frequently mis-segregate during human meiosis, producing abnormalities in chromosome number—a phenomenon termed aneuploidy. Aneuploidy is the leading cause of human pregnancy loss, as well as the cause of genetic conditions such as Klinefelter, Turner and Down syndromes[1,2]. It is estimated that only approximately half of human conceptions survive to birth, primarily because of the abundance of aneuploidies that are inviable in early gestation[5,6].

Work in humans and model organisms has established that one risk factor for aneuploidy involves variation in the number and location of meiotic crossover recombination events, especially in the female germline[3,4]. Notably, female meiosis initiates in fetal development, when replicated homologous chromosomes (homologues) pair and establish crossovers, which, together with cohesion between sister chromatids, hold homologues together in a 'bivalent' configuration. Homologues segregate (meiosis I) upon ovulation after the onset of puberty, whereas sister chromatids segregate (meiosis II) after fertilization. The physical linkages formed by meiotic crossovers help stabilize paired chromosomes during this prolonged period of female meiotic arrest[7]. Cohesin complexes, loaded in developing fetal oocytes, link sister chromatids and are crucial for chromosome synapsis and crossover formation[8,9]. Failure to form bivalents due to lack of crossovers[10] or their suboptimal placement[11], as well as age-related cohesin deterioration[12], can lead to premature separation of sister chromatids and the related phenomenon

of reverse segregation, which together represent the predominant mechanisms of maternal meiotic aneuploidy[13].

Although producing sex-specific recombination maps and revealing associations with crossover phenotypes at meiosis-related genes, the largest studies of crossovers in living human families lacked aneuploid participants and only speculated about such relationships[14,15]. Much of current knowledge about the connection between human recombination and aneuploidy, as well as their genetic bases, thus comes from smaller samples of people living with survivable aneuploidies, limiting statistical power. By contrast, recent advances in single-cell sequencing have enabled simultaneous discovery of crossovers and aneuploidies in sperm and eggs, but are typically relegated to small numbers of gametes (in the case of oocytes) or small numbers of donors, hindering understanding of variability and potential shared genetic architecture of these phenotypes[16–18].

Clinical genetic data from pre-implantation genetic testing (PGT) of in vitro fertilized (IVF) embryos help overcome these limitations and offer an ideal resource for characterizing aneuploidy and mapping meiotic crossovers at scale. Here we used single nucleotide polymorphism (SNP) array-based PGT data from 139,416 blastocyst-stage embryo biopsies and 22,850 sets of biological parents to (1) map recombination and aneuploidy, (2) test their relationship quantitatively and (3) discover genetic factors that modulate their incidence and features. Our analysis revealed an overlapping genetic basis of female recombination

[1]Department of Biology, Johns Hopkins University, Baltimore, MD, USA. [2]Department of Cellular and Molecular Medicine, DNRF Center for Chromosome Stability, University of Copenhagen, Copenhagen, Denmark. [3]Natera Inc., San Carlos, CA, USA. [4]These authors contributed equally: Sara A. Carioscia, Arjun Biddanda. ✉e-mail: rajiv.mccoy@jhu.edu

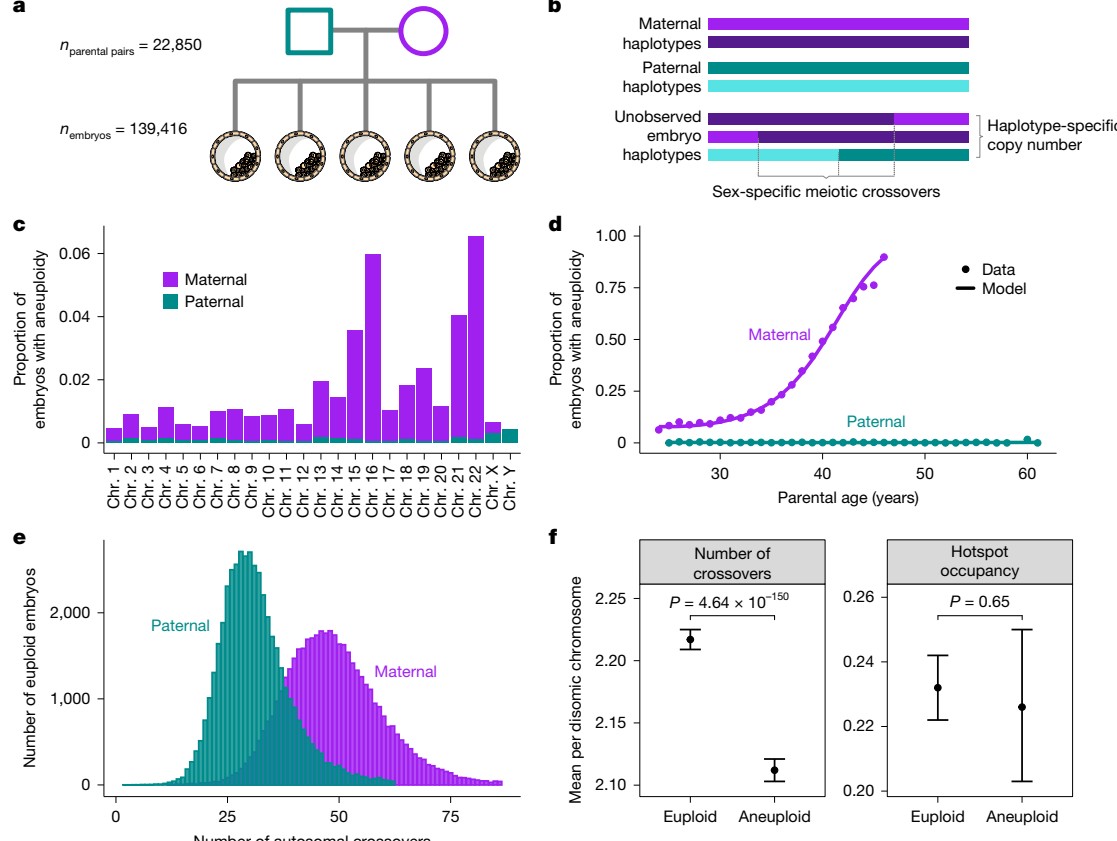

**Fig. 1 | Data from PGT of IVF embryos offer insight into crossover recombination and aneuploidy.** Colours indicate maternal (purple) versus paternal (blue) data features. **a**, Data comprise SNP microarray genotyping of trophectoderm biopsies from sibling embryos, as well as DNA from parents. **b**, Tracing transmission of parental haplotypes from parents to embryos reveals evidence of crossovers, as well as aneuploidies. **c**, Aneuploidies primarily involve gain or loss of maternal homologues and are enriched on particular chromosomes. Complex aneuploidies (more than five affected chromosomes) and genome-wide ploidy abnormalities (for example, triploidy) are excluded (Extended Data Fig. 1). **d**, Aneuploidies affecting maternal homologues increase with maternal age, whereas aneuploidies affecting paternal homologues exhibit no significant relationship with paternal age. **e**, Maternal crossovers exceed paternal crossovers. Embryos with crossover counts outside of 3 s.d. from the sex-specific mean are excluded. **f**, Crossover counts differ between disomic chromosomes of euploid ($n = 46,856$) and aneuploid ($n = 34,542$) embryos containing at least a single maternal crossover (two-sided Poisson GLMM), but the proportion of crossovers occurring within hotspots does not (two-sided Gamma GLMM). Error bars indicate 95% confidence intervals. Illustration in **a** adapted from NIH BioArt Source (https://bioart.niaid.nih.gov/bioart/209) under a Public Domain licence CC0 1.0.

and aneuploidy formation involving common variation in key meiotic machinery. Together, our work offers a more complete view of the sources of variation in the fundamental molecular processes that generate genetic diversity while impacting human fertility.

## Meiotic aneuploidy is common in embryos

Seeking insight into meiotic crossover recombination and the origins of aneuploidies, we performed retrospective analysis of data from PGT. Specifically, these data comprised SNP microarray genotyping of bulk (approximately six cells) trophectoderm biopsies from 156,828 blastocyst-stage embryos (5 days post-fertilization), as well as DNA isolated from buccal swabs or blood from both biological parents (24,788 patient–partner pairs) (Fig. 1a and Supplementary Figs. 1 and 2; Supplementary Methods). We developed a hidden Markov model (HMM), called karyoHMM, to trace the transmission of parental haplotypes to sampled embryos and thereby identify aneuploidies and crossover recombination events. Specifically, we modelled transitions between the haplotypes transmitted from the same parent as crossovers and inferred the chromosome copy number that best explained the embryo data (Fig. 1b and Supplementary Fig. 3; Supplementary Methods).

Applying this method to a dataset where low-quality samples were removed (139,416 remaining embryos; Supplementary Methods),

we identified 41,480 (29.8%) embryos with at least one aneuploid chromosome (92,485 aneuploid chromosomes; Extended Data Fig. 1). Trisomies exceeded monosomies (57,974 trisomies, 34,511 monosomies; ratio, 0.626; 95% confidence intervals, 0.624, 0.630; two-sided binomial test, $P < 1 \times 10^{-100}$), indicative of selection before blastocyst formation[6]. However, trisomies and monosomies of all individual autosomes and sex chromosomes were detected within the sample (Fig. 1c). Aneuploidies largely involved gain or loss of maternal versus paternal homologues (84,044 maternal:8,441 paternal; ratio, 0.909; 95% confidence intervals, 0.907, 0.911; two-sided binomial test, $P < 1 \times 10^{-100}$) and were strongly enriched for chromosomes 15, 16, 21 and 22, replicating previous literature[19].

We also replicated the association between maternal age and the incidence of aneuploidies affecting maternal homologues (binomial generalized linear mixed model (GLMM), $\hat{\beta} = 0.235$, s.e. $= 2.19 \times 10^{-3}$, $P < 1 \times 10^{-100}$; Supplementary Table 1)[13]. The data were well fit by a model with a quadratic term for maternal age (Fig. 1d, Supplementary Fig. 4 and Supplementary Table 1; Supplementary Methods). Positive associations with maternal age were also significant when stratifying the phenotype to maternal meiotic aneuploidy of individual chromosomes (Supplementary Table 1). Further supporting selection against meiotic aneuploidies, per patient rates of maternal meiotic aneuploidy were inversely associated with per-cycle embryo counts, even when

controlling for maternal age (binomial GLMM, $\hat{\beta} = -0.030$, s.e. $= 6.88 \times 10^{-3}$, $P = 1.29 \times 10^{-5}$). Despite the statistical power afforded by the large sample size, we observed no significant association between paternal age and aneuploidies affecting paternal homologues (binomial GLMM, $\hat{\beta} = -1.06 \times 10^{-3}$, s.e. $= 0.013$, $P = 0.936$; Fig. 1d and Supplementary Table 1), consistent with previous findings[19]. The absence of paternal age association also held for the sex chromosomes, where paternal meiotic aneuploidies were relatively more common (binomial GLMM, $\hat{\beta} = 2.14 \times 10^{-3}$, s.e. $= 0.020$, $P = 0.914$; Supplementary Table 1).

## Aneuploid embryos possess fewer crossovers

Previous studies have shown that abnormal number or placement of crossovers confers risk for meiotic aneuploidy[1,4]. These include studies of survivable trisomies[20,21], gametes[2,16,17] and embryos[16,22], which broadly demonstrated that aneuploid chromosomes are depleted of crossovers compared with corresponding disomic chromosomes.

Across 46,861 euploid embryos (and requiring at least three sibling embryos; Supplementary Methods), we identified 2,310,257 maternal- and 1,499,155 paternal-origin autosomal crossovers (3,809,412 total) at a median resolution of 99.43 kilobase pairs (kbp) (Fig. 1e). The mean counts of sex-specific crossovers per meiosis (49.30 maternal, 31.99 paternal), as well as their genomic locations (Spearman correlation ($r$) at 100-kbp resolution: 0.96 maternal, 0.98 paternal), were consistent with previous pedigree-based studies of living human cohorts[14,15]. We also observed substantial proportions of chromosomes that lack detected crossovers from a given parent (maternal = 1.67–35.56%, paternal = 7.83–51.77%), particularly among short chromosomes such as chromosomes 21 and 22 where aneuploidies are common (Extended Data Fig. 2). Acknowledging the limited resolution of the genotyping array at chromosome ends, these estimates conform with observations from living human pedigrees[14].

Previous literature offers conflicting evidence about the relationship between counts of meiotic crossovers and maternal age, with some studies reporting a positive association[14,15,23] and others reporting a negative association[24]. As those studies focused largely on living families, positive associations were interpreted typically as evidence of selection against aneuploid embryos, which possess fewer crossovers on average and increase in frequency with maternal age. Within our sample, we observed no significant association between maternal age and number of maternal crossovers (Poisson GLMM, $\hat{\beta} = -2.62 \times 10^{-5}$, s.e. $= 1.68 \times 10^{-3}$, $P = 0.988$). This observation held even when restricting analysis to euploid embryos (Poisson GLMM, $\hat{\beta} = 5.12 \times 10^{-4}$, s.e. $= 1.43 \times 10^{-3}$, $P = 0.721$), offering a point of evidence against the hypothesis that embryonic aneuploidy explains previously reported age associations with crossovers.

We used these crossover data to perform genome-wide association studies (GWAS) across four phenotypes: mean count of autosomal crossovers across euploid embryos (crossover count); fraction of crossovers within recombination hotspots based on published genetic maps (hotspot occupancy); mean timing of DNA replication at crossover sites (replication timing); and mean guanine–cytosine content ±500 bp around crossover sites (GC content; Supplementary Methods). We identified 15 unique association signals achieving genome-wide significance ($P < 5 \times 10^{-8}$), all of which replicated previous findings[14,25] (Supplementary Table 2 and Extended Data Figs. 3–6), including a haplotype spanning *RNF212* with opposing directions of association with maternal versus paternal recombination rates (lead SNP rs3816474; maternal $\hat{\beta} = -0.089 \pm 0.013$ s.e., $P = 1.84 \times 10^{-11}$; paternal $\hat{\beta} = 0.186 \pm 0.013$ s.e., $P = 1.76 \times 10^{-47}$; Extended Data Fig. 3). Complementing these GWAS, we performed transcriptome-wide association studies (TWAS) to associate predicted gene expression across several tissues[26] with recombination phenotypes, identifying 35 unique genes significantly associated with at least one recombination phenotype ($P < 3.0 \times 10^{-6}$; Supplementary Table 3; Supplementary Methods).

Prominent hits included the synaptonemal complex component *C14orf39* (also known as *SIX6OS1*)[27] and crossover-regulating ubiquitin ligase *CCNB1IP1* (also known as *HEI10*)[28], implying that previously reported genetic associations at these loci could be driven by non-coding regulatory mechanisms[14].

To examine the relationship between crossovers and aneuploidies, we contrasted patterns of crossovers between aneuploid and euploid embryos. One technical limitation for direct detection of crossovers using genetic data from trisomic chromosomes is that crossovers can be missed when both reciprocal products of a single crossover event are transmitted to the embryo[16]. To overcome this concern, we instead contrasted counts of crossovers on disomic chromosomes of aneuploid embryos (with aneuploidy affecting a different chromosome) to corresponding disomic chromosomes of euploid embryos. This comparison relies on the previous observation that crossover counts positively covary across chromosomes within meiocytes[29]—a phenomenon that we replicated for euploid embryos within our dataset (intraclass correlation coefficient = 0.176; 95% confidence intervals, 0.11, 0.3; $P < 1 \times 10^{-100}$ maternal; intraclass correlation coefficient = 0.088; 95% confidence intervals, 0.05, 0.16; $P < 1 \times 10^{-100}$ paternal; Extended Data Fig. 7; Supplementary Methods). As input to our test, we identified 1,505,107 maternal- and 1,007,176 paternal-origin crossovers on disomic chromosomes across 34,542 embryos with at least one aneuploid chromosome (and requiring at least three sibling embryos). Using a Poisson GLMM (Supplementary Methods), we found that the number of crossovers was significantly lower on the disomic chromosomes of aneuploid embryos relative to euploid embryos ($\hat{\beta} = 0.105$ difference in marginal means $\pm 6.923 \times 10^{-5}$ s.e.; $P = 4.64 \times 10^{-150}$; Fig. 1f). These results are consistent with the understanding that reduction in crossovers—and absence of crossovers, in particular[10]—confers risk for meiotic aneuploidy.

## *SMC1B* variants associate with aneuploidy

Previous studies have suggested that the incidence of female meiotic aneuploidy may be individual-specific, even after accounting for maternal age[30]. To test this hypothesis, we fit a quasi-binomial generalized linear model (GLM) to the per patient counts of embryos affected versus unaffected with maternal meiotic-origin aneuploidy, including maternal age as a quadratic covariate (Supplementary Methods). Compared with a simulated binomial null distribution, the observed incidence of meiotic aneuploidy was significantly overdispersed across female patients, controlling for maternal age (dispersion parameter ($\varphi$) = 1.15, $P < 0.01$; Supplementary Fig. 5). Overdispersion was also apparent when stratifying analysis to maternal meiotic aneuploidies affecting individual chromosomes (Supplementary Table 4). These observations of overdispersion suggest a role of genetic and environmental factors beyond age in observed variation in maternal meiotic aneuploidy.

To investigate the genetic component, we scanned for variation in maternal genomes associated with the incidence of maternal meiotic aneuploidy. We implemented these association tests using a binomial GLMM, controlling for covariates including maternal age (Supplementary Methods). We first tested for *cis*-genetic effects on aneuploidy risk by associating incidence of aneuploidy affecting each individual chromosome with maternal genotypes restricted to that chromosome, but we identified no associations achieving genome-wide significance ($P < 5 \times 10^{-8}$). Proceeding to a genome-wide analysis considering maternal meiotic aneuploidies affecting any chromosome, we discovered two genome-wide significant associations (Fig. 2a and Supplementary Fig. 6). The first hit (lead SNP rs9351349, $\hat{\beta} = 0.078$, s.e. $= 0.014$, $P = 2.93 \times 10^{-8}$) lies within an intergenic region of chromosome (Chr.) 6 but did not replicate in a held-out test set comprising 15% of female patients ($\hat{\beta} = 0.021$, s.e. $= 0.033$, $P = 0.529$). The second hit (lead SNP rs6006737, $\hat{\beta} = 0.066$, s.e. $= 0.012$, $P = 2.21 \times 10^{-8}$) lies on Chr. 22 and replicated in the held-out test set ($\hat{\beta} = 0.059$, s.e. $= 0.028$, $P = 0.033$).

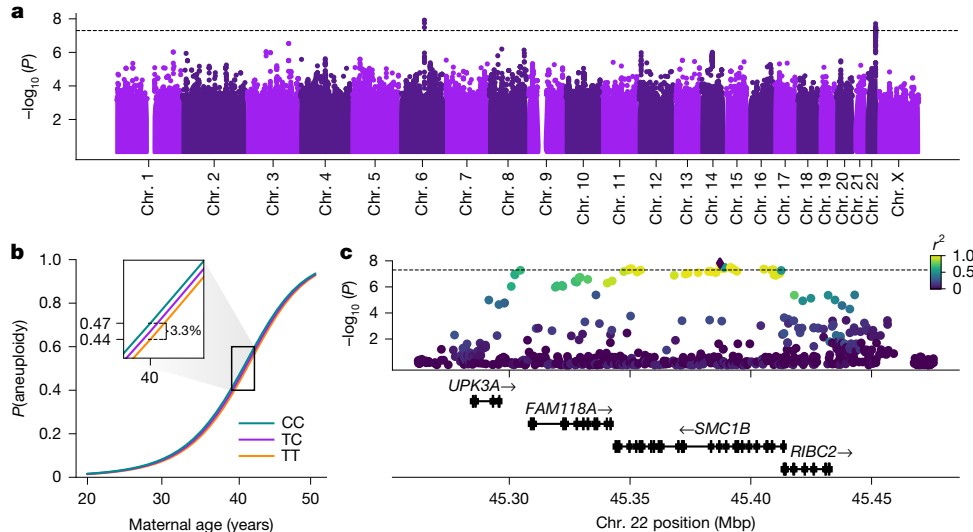

**Fig. 2 | Variants defining a haplotype spanning *SMC1B* are associated with incidence of maternal meiotic aneuploidy. a**, Results of GWAS tests of maternal meiotic aneuploidy and maternal genotype (two-sided binomial GLMM). The dotted line indicates the threshold for genome-wide significance ($P = 5 \times 10^{-8}$). **b**, Fitted relationship between maternal age and incidence of aneuploidy, stratified by maternal genotype at aneuploidy-associated lead SNP rs6006737. **c**, Regional Manhattan plot depicting the associated locus on Chr. 22, with points coloured based on pairwise linkage disequilibrium with the lead SNP rs6006737 (diamond). Mbp, megabase pairs.

The minor (C) allele of rs6006737 within our sample is globally common, segregating at high frequencies (gnomAD allele frequency (AF) = 0.78) in African populations but at lower frequencies in European (gnomAD AF = 0.35) and other non-African populations[31]. The effect is additive, whereby for a 40-year-old patient, each copy of the risk allele confers an estimated 1.65% additional average risk of aneuploidy (Fig. 2b). We also detected evidence of a small but statistically significant interaction between maternal age and genotype (likelihood ratio test, $\chi^2(1) = 4.24$, $P = 0.040$), indicating that the effect of genotype increases with increasing maternal age ($\hat{\beta} = 0.026$, s.e. = 0.013, $P = 0.045$). Notably, the size and direction of the main effect of genotype is relatively consistent for aneuploidies of all individual autosomes (Extended Data Fig. 8), suggesting general, genome-wide impacts on meiotic fidelity.

The associated haplotype spans approximately 120 kbp, encompassing four genes: *UPK3A*, *FAM118A*, *RIBC2* and *SMC1B* (Fig. 2c). *SMC1B* encodes a component of the ring-shaped cohesin complex (Fig. 3a), with integral roles in sister chromatid cohesion and homologous recombination during meiosis[32,33]. *Smc1b*-deficient mice of both sexes are sterile, and female mice exhibit meiotic abnormalities including reduction in crossovers, incomplete chromosome synapsis, age-related premature loss of sister chromatid cohesion and chromosome missegregation[32,33]. Previous work in humans demonstrated associations between a less common (gnomAD global AF = 0.06) *SMC1B* missense variant (rs61735519; $r^2$ with GWAS lead SNP rs6006737 = 0.089, $D' = 0.943$) and recombination phenotypes[14]. Although imputed with moderate accuracy (dosage $r^2 = 0.80$), this missense variant exhibits only modest association with aneuploidy within our sample ($\hat{\beta} = 0.112$, s.e. = 0.040, $P = 4.80 \times 10^{-3}$). Meanwhile, the more common aneuploidy-associated haplotype tagged by GWAS lead variant rs6006737 lacks amino acid altering variation ($r^2 < 0.1$ for all *SMC1B* nonsynonymous variants), motivating us to explore potential regulatory mechanisms driving the observed phenotype.

## Associated haplotype is an *SMC1B* expression quantitative trait locus

Querying the GWAS lead variant (rs6006737) in data from the Genotype Tissue Expression (GTEx) Project[26], we observed that the aneuploidy risk allele is associated significantly with reduced expression of *SMC1B* across diverse tissues. Although invaluable, GTEx largely includes

participants of European ancestries, limiting resolution for fine-mapping of causal expression-altering variants. To address this limitation, we also queried the GWAS lead variant in MAGE, which includes RNA sequencing data from lymphoblastoid cell lines from 731 people from 26 globally diverse populations[34]. Consistent with GTEx, rs6006737 is a strong expression quantitative trait locus (eQTL) of *SMC1B* in MAGE ($\hat{\beta} = -0.429$, s.e. = 0.048, $P = 4.68 \times 10^{-18}$; Fig. 3b). Fine-mapping within MAGE decomposes the eQTL signals for *SMC1B* into two credible sets containing candidate causal variants (coverage = 0.95) (Fig. 3c,d). Whereas one credible set includes nine variants distributed throughout the upstream region of *SMC1B*, the other is defined by a single SNP (rs2272804; posterior inclusion probability > 0.99), 144 bp upstream of the *SMC1B* transcription start site.

The regulatory potential and accessibility of the putative promoter CpG island sequence within which rs2272804 resides is supported by published epigenomic and ATAC-seq (assay for transposase-accessible chromatin using sequencing) data from human ovaries[35,36] (Fig. 3d). We further noted that the SNP lies within a predicted binding motif of ATF1—a transcription factor expressed in female germ cells[37] and inferred previously to regulate paralogue *SMC1A* based on chromatin immunoprecipitation sequencing data[38]. Binding of ATF1 to the SNP-encompassing locus is also supported by high-confidence chromatin immunoprecipitation sequencing peaks in induced pluripotent stem cells (WTC11) assayed by the ENCODE Project[38]. By performing an electrophoretic mobility shift assay, we found that a DNA construct containing the alternative allele of rs2272804 had more than three-fold lower binding affinity (dissociation constant, $K_D$) for purified human ATF1 in vitro than a construct containing the reference allele (Student's *t*-test, mean reference $K_D = 56.62$ nM ± 4.65 s.d., mean variant $K_D = 173.39$ nM ± 15.24 s.d., $P = 2.60 \times 10^{-4}$), consistent with the observed eQTL effect (Extended Data Fig. 9). Taken together, these results suggest a potential non-coding regulatory mechanism underlying the observed genetic association with maternal meiotic aneuploidy.

## TWAS reveals new links to meiosis genes

Motivated by our observations at *SMC1B*, we next sought to examine whether other *cis*-regulatory effects on expression could influence aneuploidy risk. We therefore used TWAS to test whether predicted gene expression across tissues is associated with incidence of aneuploidy

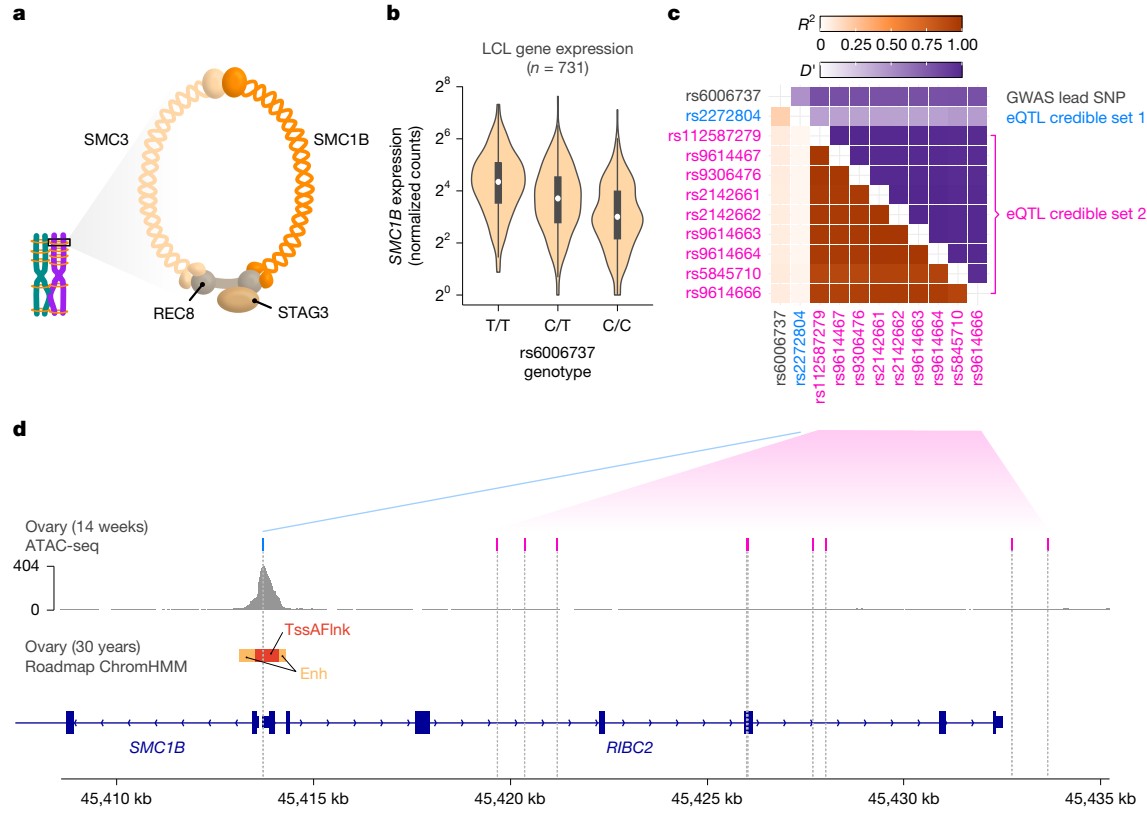

**Fig. 3 | The aneuploidy risk haplotype is associated with lower expression of *SMC1B*, driven by two independent causal signals. a**, Schematic of the meiotic cohesin complex. **b**, Each copy of the aneuploidy risk allele is associated with reduced expression of *SMC1B* in human lymphoblastoid cell lines (LCLs; *n* = 731) from diverse populations. Bars represent the first and third quartiles of the data, white points represent the second quartile (median) and whiskers are bound to 1.5× the interquartile range. **c**, Pairwise linkage disequilibrium between a set of SNPs including GWAS lead SNP rs6006737 and variants defining fine-mapped eQTL credible sets for *SMC1B*. **d**, Fine-mapped eQTL rs2272804 (credible set 1) lies within a putative promoter sequence within open chromatin, while variants defining a second credible set are distributed throughout the upstream region of *SMC1B*.

(Supplementary Methods). Across 16,685 protein-coding genes, we identified two hits achieving transcriptome-wide significance ($P < 3 \times 10^{-6}$; Extended Data Fig. 10). Although led by adjacent gene *RIBC2* ($P = 2.19 \times 10^{-7}$), the peak on Chr. 22 includes *SMC1B* ($P = 7.63 \times 10^{-6}$), replicating our findings from GWAS and downstream functional dissection. We hypothesize that *RIBC2* represents a secondary, noncausal association, whereby the same haplotype (and potentially the same causal variant[39]) co-regulates expression of both genes, driving their correlation (Supplementary Fig. 7). The second peak lies on Chr. 14 and is led by *C14orf39* ($P = 1.65 \times 10^{-7}$), which encodes a component of the synaptonemal complex, which mediates synapsis, recombination and segregation of homologous chromosomes during meiosis[27]. Previous studies have linked rare *C14orf39* variants to human infertility[40,41] and demonstrated associations between common *C14orf39* variants and recombination phenotypes[14,25]. Our results connect these findings and show that both rare and common variation influencing female fertility phenotypes can converge on the same meiosis-related genes. Although not achieving transcriptome-wide significance, a third peak, on Chr. 12, includes *NCAPD2* ($P = 2.16 \times 10^{-5}$), which encodes a regulatory subunit of the condensin I complex, involved in chromosome condensation during both meiotic and mitotic prophase[42]. Together, our findings highlight the role of common non-coding *cis*-regulatory variation influencing expression of meiosis-related genes in modulating risk of maternal meiotic aneuploidy (Extended Data Fig. 10).

## Pleiotropic effects on fertility traits

Given the relationship between crossovers and aneuploidies, we next aimed to contextualize our association findings and examine the potential shared genetic basis with other fertility-related traits. To this end, we identified the lead variant from each genome-wide significant peak in female recombination and aneuploidy GWAS and queried their associations with all recombination and aneuploidy phenotypes, as well as published GWAS of female reproductive ageing and infertility traits (that is, phenome-wide association). Our analysis revealed that the risk allele of the aneuploidy-associated lead SNP rs6006737 is also associated with lower rates of female recombination within our data ($\hat{\beta} = -0.033$, s.e. = 0.011, $P = 0.002$), consistent with the known role of *SMC1B* variation in this phenotype[32]. Extending to published GWAS data[43,44], we observed that the aneuploidy risk allele is additionally associated with greater age at menarche ($\hat{\beta} = 0.021$, s.e. = 0.003, $P = 3.82 \times 10^{-12}$) and lesser age at menopause ($\hat{\beta} = -0.047$, s.e. = 0.013, $P = 2.06 \times 10^{-4}$) and thus a shorter female reproductive timespan (Fig. 4).

Strikingly, three of the genome-wide significant hits for female recombination rate (Supplementary Table 2) also exhibited nominal associations with aneuploidy in consistent direction. The first hit (lead SNP rs4365199; aneuploidy $\hat{\beta} = 0.056$, s.e. = 0.012, $P = 5.58 \times 10^{-6}$; gnomAD global AF = 0.39) comprises a 175-kbp haplotype spanning synaptonemal complex component *C14orf39*, consistent with our previous TWAS results. The second hit (lead SNP rs12588213; aneuploidy $\hat{\beta} = 0.037$, s.e. = 0.012, $P = 1.46 \times 10^{-3}$; gnomAD global AF = 0.42) comprises a 15-kbp haplotype spanning *CCNB1IP1*, encoding an E3 ubiquitin ligase demonstrated as essential for crossover maturation and fertility in mice[28]. The last hit (lead SNP rs3816474; aneuploidy $\hat{\beta} = 0.041$, s.e. = 0.014, $P = 5.04 \times 10^{-3}$; gnomAD global AF = 0.22) comprises a 59-kbp haplotype spanning the E3 ubiquitin ligase *RNF212*, encoding an essential regulator of meiotic recombination that interacts with *CCNB1IP1* and helps to designate sites of crossovers versus

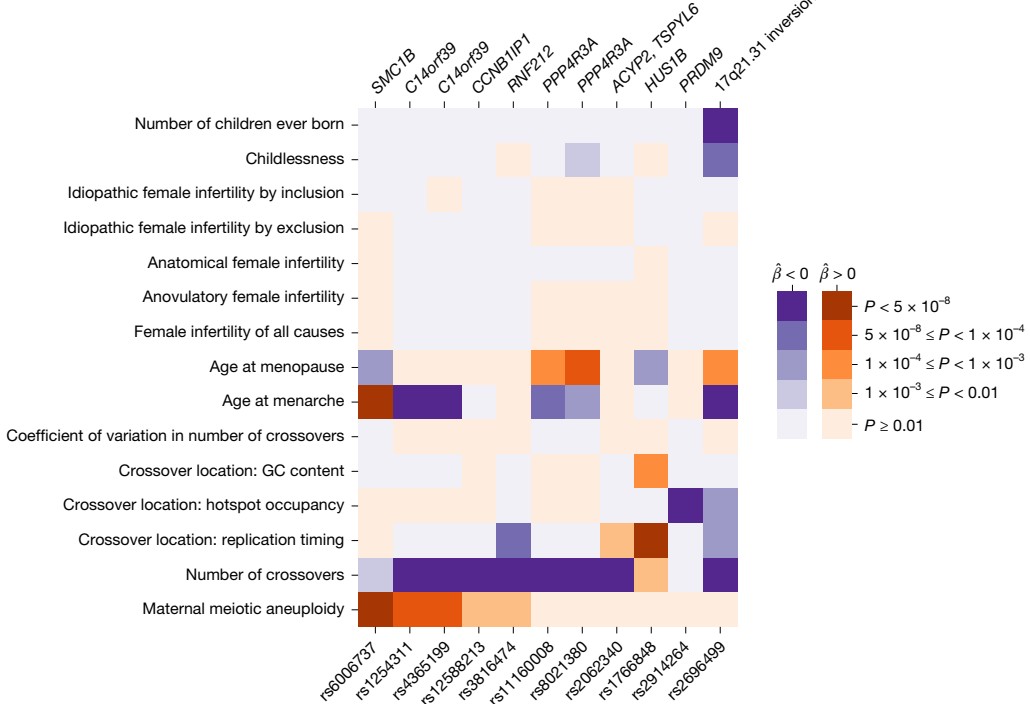

**Fig. 4 | Aneuploidy, recombination and female reproductive ageing traits share an overlapping genetic basis.** The lead SNP from each peak from GWAS of aneuploidy and recombination was queried for association with other fertility-related phenotypes (two-sided linear or logistic model from respective GWAS study). Darkness indicates significance of association ($P$ value), while colour indicates direction of association. SNPs are polarized such that the aneuploidy-increasing allele is queried across all traits. Each hit is labelled based on meiosis-related candidate genes within the associated region (top) with the exception of the common 17q21.31 inversion, as well as the locus containing *ACYP2* and *TSPYL6*, where no such candidate is apparent.

non-crossovers[45]. Several of these recombination and aneuploidy-associated variants also exhibited secondary associations with ages at menarche and menopause (Fig. 4). Whereas previous studies have reported links between DNA damage response and reproductive ageing[43,46,47], the inconsistencies in directions of effects in our data imply that the relationship with aneuploidy may be more complex. Moreover, none of the aneuploidy-associated variants exhibited even nominal associations with various definitions of female infertility[48], potentially reflecting the multifactorial nature of clinical infertility.

Despite our discoveries of several genome- and transcriptome-wide significant loci, the proportion of variance in maternal meiotic aneuploidy explained by genotyped SNPs (that is, SNP heritability) was negligible ($h^2_{SNP} = 0.023 \pm 0.024$ s.e.; Supplementary Table 5), although SNP heritability of female recombination rate was moderately higher ($h^2_{SNP} = 0.112 \pm 0.042$ s.e.). These estimates are in line with low reported SNP heritabilities of female fertility phenotypes[48] and the sizeable contribution of environmental factors to maternal aneuploidy risk. Given these observations, we hypothesized that environmental factors and/or rare genetic variation contribute to residual variance in aneuploidy rates, including by effects on meiotic recombination. In support of this hypothesis, individual-specific rates of recombination were inversely associated with aneuploidy, even after controlling for maternal age and all genetic associations (binomial GLMM, $\hat{\beta} = -0.763$, s.e. $= 0.14$, $P = 8.15 \times 10^{-8}$; Supplementary Methods), again supporting a broad, protective effect of crossovers on aneuploidy risk.

## Evolution of the *SMC1B* risk allele

The discovery of a common aneuploidy-associated haplotype at *SMC1B* poses an evolutionary paradox, as alleles that reduce fitness should be subject to negative natural selection. To understand the evolution of aneuploidy-associated alleles, we examined empirical signatures of natural selection and explored the theoretical parameter space that would allow us to reconcile these observations.

One potential model for explaining the maintenance of deleterious variation is positive or balancing selection targeting the same haplotype. Given that linkage disequilibrium between causal variants and tagging variants differs across populations and over time, we focused our empirical analyses on the putative causal expression-altering SNP, rs2272804, that we previously characterized. The (A) allele of rs2272804, associated with lower *SMC1B* expression and higher aneuploidy risk, is globally common (gnomAD global AF = 0.44), with higher frequencies among African populations (gnomAD AF = 0.71). Inference of the historical frequency trajectory of the derived risk allele based on the ancestral recombination graph (Supplementary Methods) also suggests a higher frequency within an ancestral human population, modestly declining outside of Africa within the last 1,000 generations (Fig. 5a). While the putative ancestral (C) allele appears fixed among extant non-human great ape populations, the variant is polymorphic across high-coverage Neanderthal genomes (Supplementary Methods), and coalescence-based methods estimate that the derived allele originated 910,650 years ago (95% confidence intervals, 825,825–1,004,175)[49]. These patterns of frequency differentiation and coalescence are unremarkable and broadly conform to neutral expectations for a variant at such intermediate frequencies[49]. Similarly, haplotype-based tests for balancing selection revealed no outlier signal in the region of *SMC1B* (Supplementary Fig. 8; Supplementary Methods). Although we cannot formally exclude the possibility of more complex histories or subtle signatures of positive or balancing selection at this locus, we next considered a theoretical model of negative selection.

Specifically, we formulated a mathematical model (Supplementary Note 1) that integrates over the maternal reproductive timespan and contrasts the potential lifetime production of chromosomally

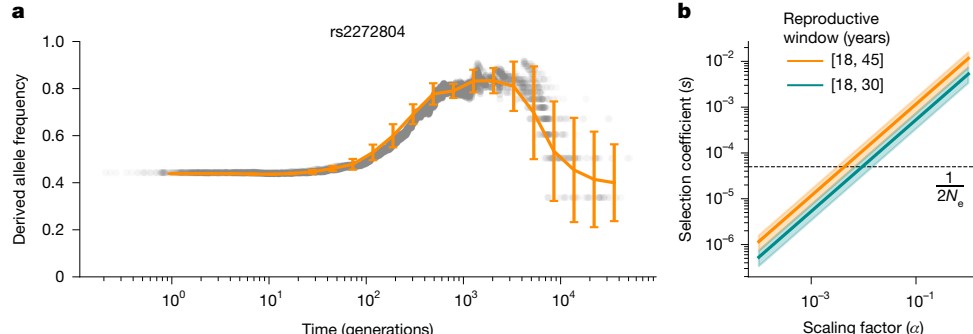

**Fig. 5 | Evolutionary modelling of maternal meiotic aneuploidy risk haplotype. a**, Estimated allele frequency trajectory of the derived (C) allele at rs2272804 inferred from 100 posterior samples of the ancestral recombination graph at the *SMC1B* locus. Posterior mean allele frequencies and s.e. were computed within log-spaced time bins, and error bars indicate ±2 s.e. around the posterior mean. **b**, Relationship between the scaling factor ($\alpha$) relating fitness to the fitness proxy (number of embryos lacking maternal meiotic aneuploidies) and the selection coefficient ($s$) for two illustrative female reproductive time windows (Supplementary Note 1). Shaded areas denote 95% confidence intervals of the estimated effect of the haplotype tagged by GWAS lead variant rs6006737 on the resultant mean scaling factor. The horizontal dashed line indicates the theoretical threshold for neutral evolution ($\frac{1}{2N_e}$), assuming a human effective population size of $10^4$.

normal embryos between carriers and non-carriers of the aneuploidy risk allele. The ratios of these proxies for relative fitness can be used to derive a proxy for the selection coefficient ($s_{proxy}$). Based on this model, we estimated that for a historical maternal reproductive window between 18 and 35 years of age, $s_{proxy} \approx 0.01$ and increases moderately upon increasing the upper bound of maternal age (Fig. 5b and Supplementary Fig. 9). For a human effective population size ($N_e$) on the order of $10^4$, a selection coefficient of 0.01 is much greater than the theoretical threshold of $\frac{1}{2N_e}$, implying that the allele should be subject to negative selection.

However, although the number of euploid embryos a woman can produce is presumably correlated with fitness, it may constitute only a weak proxy, as realized fitness is also determined by stochastic, environmental and behavioural factors largely independent of genotype. Moreover, pregnancy, childbirth and miscarriage can influence fitness in complex ways, including through impacts on maternal survival, future fertility and parental/grandparental care[50]. We observed that to reach the theoretical threshold for evading negative selection, the selection coefficient ($s$) must be scaled by a factor ($\alpha$) ≤ 0.01, relative to $s_{proxy}$ (Supplementary Note 1). Although the historical relationship between fitness and the fitness proxy is unknown, their weak correlation in contemporary populations is evidenced by the lack of association between the aneuploidy risk variant and fertility phenotypes such as number of children ever born and childlessness[51] (Fig. 4). These results highlight the inadequacy of simplistic proxies of fitness—a limitation long appreciated in the field of life history theory[52]—while reconciling the observation of a common aneuploidy-associated allele.

## Discussion

Pregnancy loss is common in humans[5] and often traces to aneuploidy originating in the maternal germline[1]. Notably, female meiosis initiates in fetal development, when homologous chromosomes pair and establish crossovers, but arrests for decades until ovulation and fertilization. Abnormal number or placement of crossovers predisposes oocytes to chromosome mis-segregation upon meiotic resumption[4,10]. Despite this understanding, the role of common genetic variation in modulating these important molecular processes in humans has remained poorly understood. Through retrospective analysis of large-scale PGT data from human IVF embryos, we mapped genetic variants associated with crossover and aneuploidy phenotypes, revealing an overlapping genetic basis involving key meiosis genes.

Although we measured overdispersion in the age-adjusted rate of aneuploidy per patient and identified genome- and transcriptome-wide

significant associations, we were intrigued to find that the SNP heritability of aneuploidy was negligible. This finding aligns with low reported SNP heritabilities of female infertility phenotypes[48], as well as potential outsize contributions of environmental and rare genetic variation influencing this trait. Nevertheless, given that common and rare variation often converge on the same genes and mechanisms[53], our results may help inform sequencing-based studies of aneuploidy phenotypes. Supporting a model of mechanistic convergence, rare loss-of-function mutations in several of the genes implicated here have also been linked to meiotic defects and reproductive disorders in smaller clinical cohorts[40,54]. It is also plausible that a fraction of phenotypic variance for aneuploidy risk could trace to common genetic variation that is inaccessible to genotyping arrays and/or short-read sequencing, for example within technically challenging loci such as large segmental duplications, telomeres or centromeres. Recent work offered preliminary evidence that particular centromeric haplotypes are enriched among cases of Trisomy 21 (ref. 55). Future applications of long-read sequencing in PGT may enable validation of this hypothesis and extension to inviable aneuploidies.

The observation that alleles associated with lower rates of recombination are associated with higher rates of aneuploidy raises interesting questions about the evolutionary forces that shape recombination and aneuploidy within and between species. In addition to generating new combinations of alleles, recombination may also induce point mutations and structural variation near hotspots of double-strand breaks[14,56]. This suggests a model of stabilizing selection, whereby rates of recombination may be constrained on the lower and upper ends to limit aneuploidy and other deleterious mutations, respectively. More comprehensive models of recombination rate evolution must also consider mechanical constraints such as crossover interference, which reduces occurrence of nearby crossovers, as well as the role of crossovers in facilitating adaptation. By examining divergence across a mammalian phylogeny, a recent study reported signatures of pervasive positive selection on all meiotic components of the cohesin complex (*SMC1B*, *RAD21L1*, *REC8* and *STAG3*), which the authors speculated could be explained by intragenomic conflict[57]. Although the asymmetry of female meiosis is susceptible to meiotic drive, the role of meiotic drive in the origins of human aneuploidy remains an important open question.

More broadly, the observation that common genetic variants modulate key reproductive phenotypes such as aneuploidy and recombination poses an intriguing evolutionary paradox, as theory predicts that variation that strongly reduces fitness should be subject to negative selection. We present a theoretical model of negative selection that interprets GWAS effects in terms of potential lifetime production of

viable embryos. Our model places an upper bound on the strength of the relationship between this fitness proxy and realized fitness that would allow risk alleles to evade negative selection and reach intermediate frequencies by genetic drift. This framework could be generalized to guide expectations for future studies examining the genetic architecture of aneuploidy and other fertility-related traits.

In summary, our work provides a more complete understanding of common genetic factors that influence risk of aneuploidy—the leading cause of human pregnancy loss. These findings highlight the interplay among the forces of mutation, recombination and natural selection that operate before birth to shape human genetic diversity.

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

## Methods

Detailed methods are provided in the Supplementary Information.

### Ethics statement

The Johns Hopkins University Homewood Institutional Review Board (IRB) determined that this research did not qualify as federally regulated human participants research and therefore did not require IRB approval (HIRB00011705). This determination was made with the understanding that the research (1) does not involve a systematic research investigation designed to develop or contribute to generalizable knowledge, or (2) does not obtain information or biospecimens through intervention or interaction with a human participant, and use, study or analyse the information or biospecimens; or does not obtain, use, study, analyse or generate identifiable private information or identifiable biospecimens. Data collection and analysis was carried out in compliance with Natera's IRB approved protocol (Salus no. 10806) involving Category 4 Exempt Research.

### Reporting summary

Further information on research design is available in the Nature Portfolio Reporting Summary linked to this article.

### Data availability

Association study summary statistics and aneuploidy and crossover calls are available at Zenodo (https://doi.org/10.5281/zenodo.15114527)[58]. Patient privacy and data use restrictions do not allow for external deposition of raw SNP microarray data. Researchers may request access to these data from Natera by contacting Zachary Demko (zdemko@natera.com). Initial responses to these requests can be expected within 1 month of receipt. Approved requests will require separate data use agreements in compliance with appropriate privacy laws. Publicly available resources used in the study include: GRCh37 and GRCh38 (https://hgdownload.soe.ucsc.edu/), GENCODE v.37 (https://www.gencodegenes.org/human/release_37.html), gnomAD (https://gnomad.broadinstitute.org/), MAGE (https://github.com/mccoy-lab/MAGE/), GTEx v.8 (https://github.com/broadinstitute/gtex-v8), ENCODE (https://www.encodeproject.org/), deCODE genetic map (https://www.science.org/doi/10.1126/science.aau1043; Supplementary File 1), and the 1000 Genomes Project + HGDP reference panel (https://github.com/atgu/hgdp_tgp).

### Code availability

Genotyping and imputation code is available at GitHub (https://github.com/mccoy-lab/natera_genotyping/) and Zenodo (https://doi.org/10.5281/zenodo.17429676)[59]. Pipelines for inferring crossover recombination across sibling embryos are available at GitHub (https://github.com/mccoy-lab/natera_recomb) and Zenodo (https://doi.org/10.5281/zenodo.17429678)[60]. Code for inferring aneuploidies and performing downstream analyses is available at GitHub (https://github.com/mccoy-lab/karyohmm; https://github.com/mccoy-lab/natera_aneuploidy) and Zenodo (https://doi.org/10.5281/zenodo.17429669; https://doi.org/10.5281/zenodo.17429672)[61,62].

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

**Acknowledgements** We thank all participants who contributed data used in this study. We also thank Advanced Research Computing at Hopkins for computing support, as well as C. Wu, E. Andersen, Y. Kim, Y. He, D. Petrov, E. Koch, N. Chatterjee, K. Schindler, J. Xing, G. Taumoefolau and members of the McCoy laboratory and Origins of Aneuploidy Research Consortium for helpful input. We thank G. Gemelos and D. Kijacic for assistance with data collection. This work is supported by a National Science Foundation Graduate Research Fellowship (1746891) to S.A.C., a Lalor Foundation Postdoctoral Fellowship to A.B., a National Institutes of Health (NIH NIGMS) grant R35GM149291 to C. Wu, a Novo Nordisk Foundation grant NNF22OC0074308 to E.R.H., Catalyst and Discovery Awards from Johns Hopkins University to R.C.M. and a National Institutes of Health (NIH NIGMS) grant R35GM133747 to R.C.M. The content is solely the responsibility of the authors and does not necessarily represent the official views of the funders.

**Author contributions** R.C.M. initially conceived the project with input from E.R.H., A.B. and S.A.C. Z.P.D. contributed to collection of the data. Algorithms for inferring aneuploidy and recombination were developed by A.B. with input from S.A.C. Parental genotyping and imputation were performed by S.A.C. and A.B. Statistical modelling of recombination was performed by R.C.M. and A.B. Recombination GWAS and evolutionary modelling were performed by A.B. Aneuploidy GWAS, functional genetic analysis of association hits, and phenome-wide association analyses were performed by S.A.C. TWAS was performed by M.R.S. Electrophoretic mobility shift assay was performed by X.T. S.A.C., A.B. and R.C.M. wrote the paper and generated the figures. All authors read and approved the manuscript.

**Competing interests** Z.P.D. is a full-time employee and holds stock or options to hold stock in Natera, Inc. E.R.H. is a co-founder of Ovartix, Ltd. R.C.M. is a co-inventor of a method for analysis of PGT-A data which is the subject of a US patent (12,322,509) assigned to Johns Hopkins University; this method was not applied in the present study. The other authors declare no competing interests.

**Additional information**
**Correspondence and requests for materials** should be addressed to Rajiv C. McCoy.

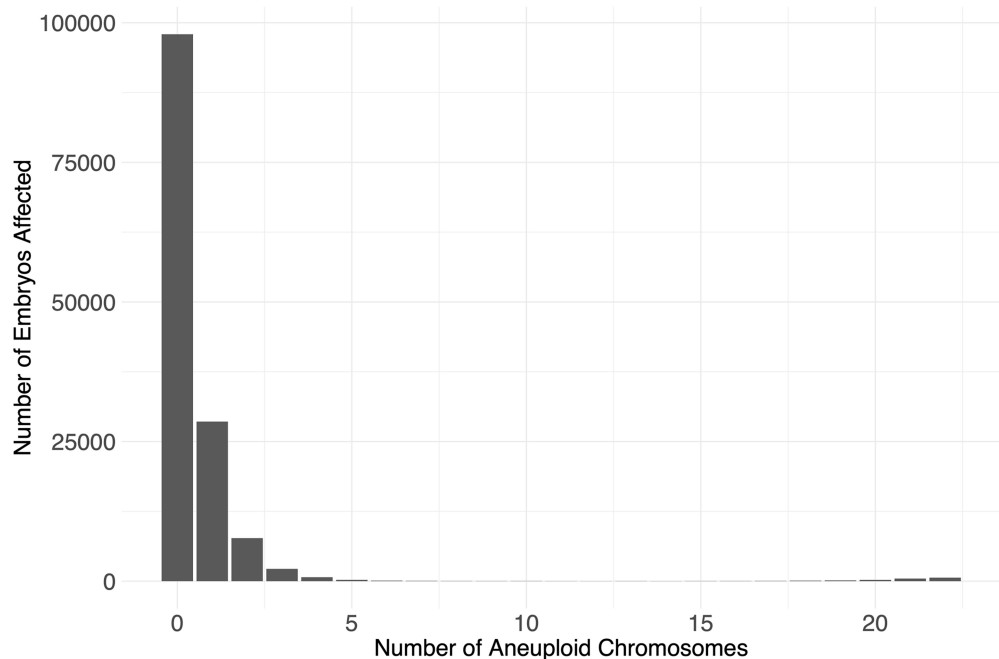

**Extended Data Fig. 1 | Distribution of aneuploid chromosomes across embryos.** Of all tested embryos, 70.25% were called as euploid, 20.49% exhibited aneuploidy affecting a single chromosome, and 9.26% exhibited more complex abnormalities affecting multiple chromosomes, including 1.08% exhibiting whole-genome gain and 0.23% exhibiting whole-genome loss or genome-wide uniparental isodisomy.

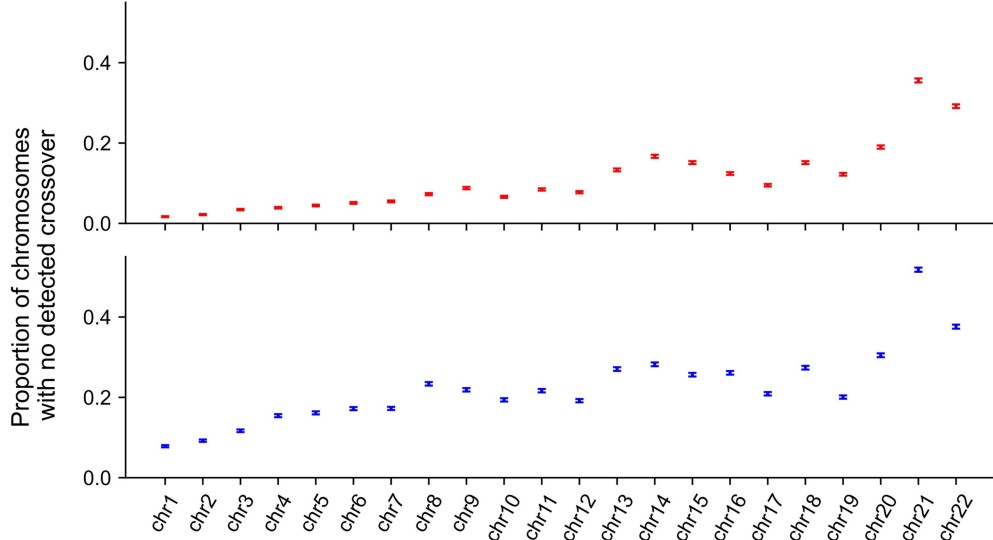

**Extended Data Fig. 2 | Proportions of chromosomes lacking detected crossovers from a given parent.** Data are restricted to euploid embryos and depict the proportion of embryos without any observed crossover for each chromosome, stratified by maternal (top) and paternal (bottom) crossovers.

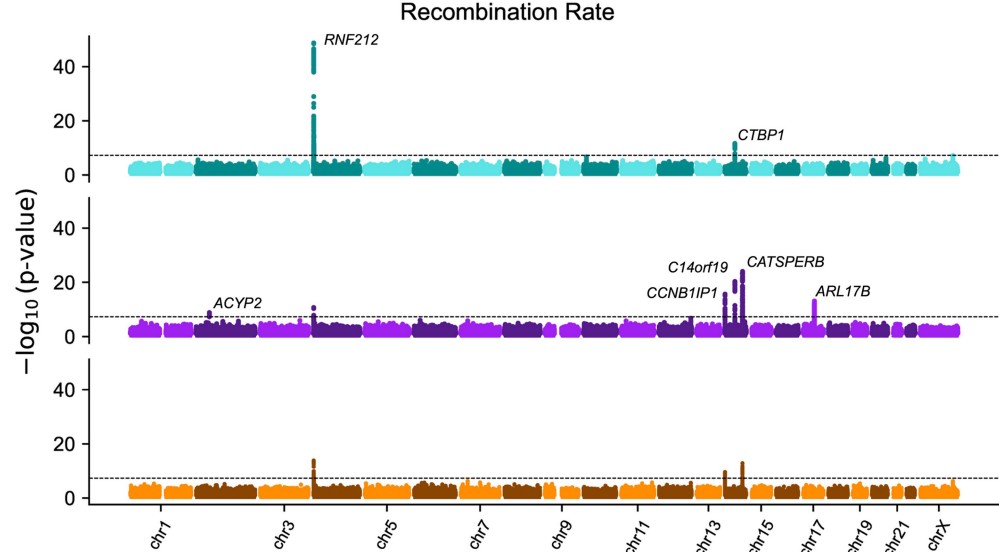

**Extended Data Fig. 3 | Manhattan plots for GWAS of recombination rate.** Results per trait are stratified by paternal (cyan), maternal (purple) and joint (orange) analyses. Dashed line reflects the genome-wide significance threshold ($p < 5 \times 10^{-8}$). Physically nearest genes to lead GWAS variant per-peak are annotated per trait.

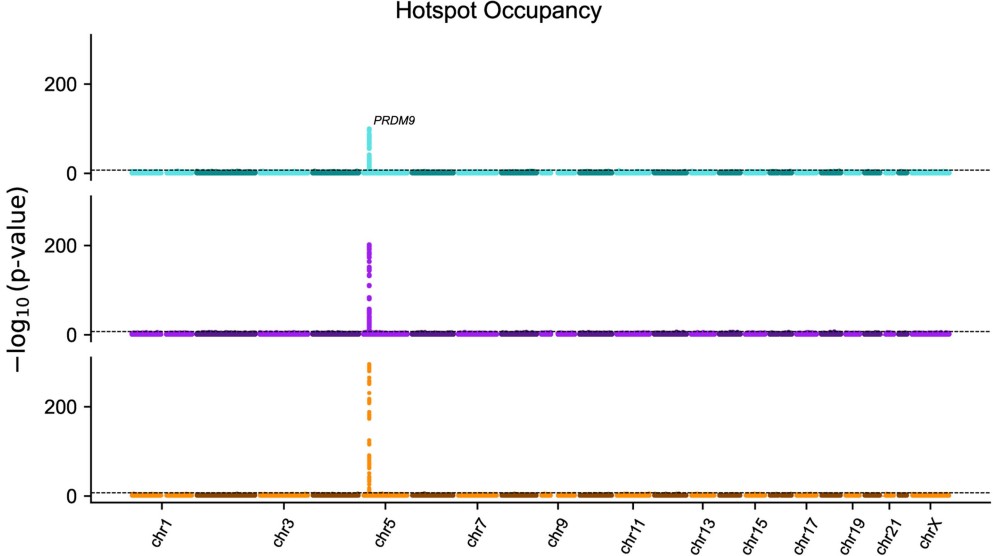

**Extended Data Fig. 4 | Manhattan plots for GWAS of recombination hotspot occupancy.** Results per trait are stratified by paternal (cyan), maternal (purple) and joint (orange) analyses. Dashed line reflects the genome-wide significance threshold ($p < 5 \times 10^{-8}$). Physically nearest genes to lead GWAS variant per-peak are annotated per trait.

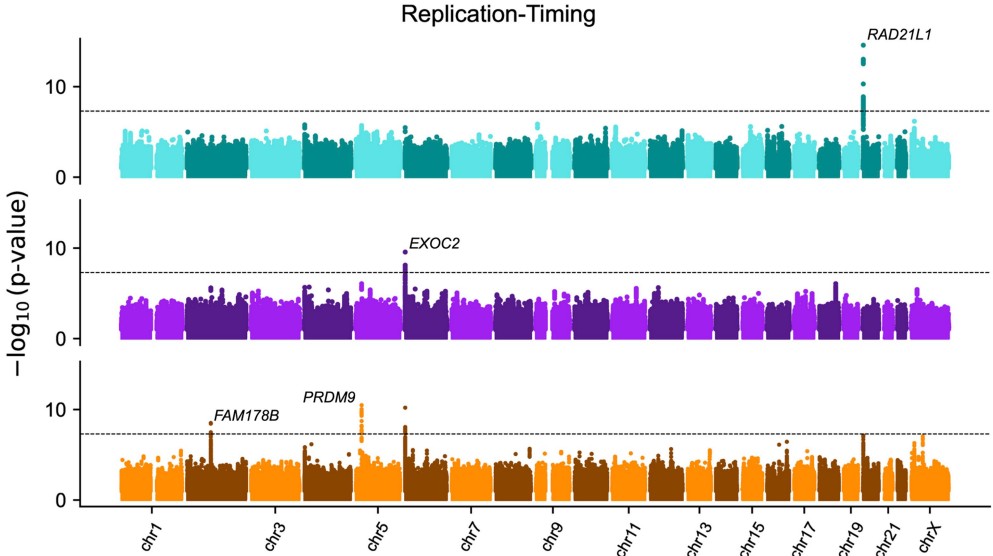

**Extended Data Fig. 5 | Manhattan plots for GWAS of estimated replication timing at hotspots.** Results per trait are stratified by paternal (cyan), maternal (purple) and joint (orange) analyses. Dashed line reflects the genome-wide significance threshold ($p < 5 \times 10^{-8}$). Physically nearest genes to lead GWAS variant per-peak are annotated per trait.

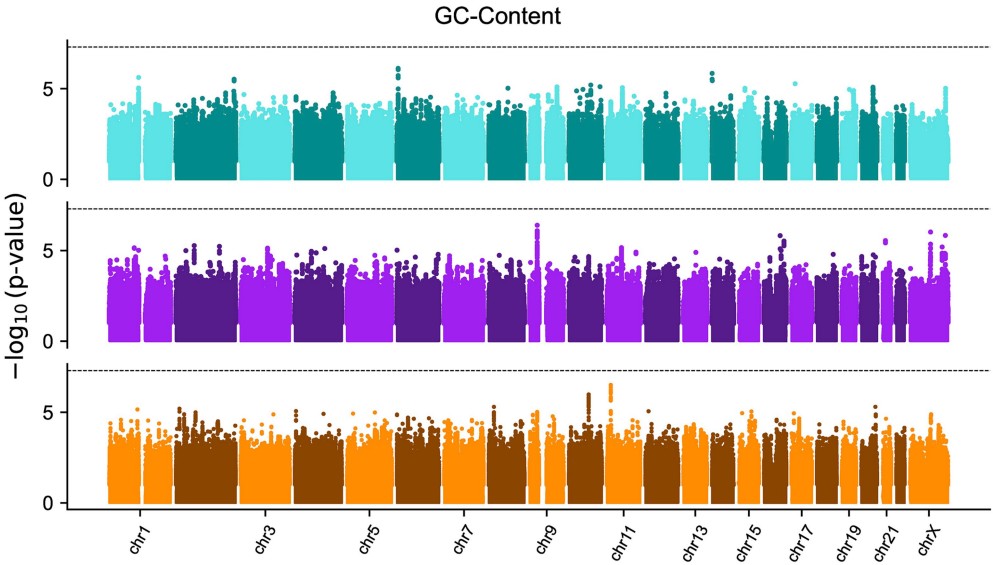

**Extended Data Fig. 6 | Manhattan plots for GWAS of GC content at crossover locations.** Results per trait are stratified by paternal (cyan), maternal (purple) and joint (orange) analyses. Dashed line reflects the genome-wide significance threshold ($p < 5 \times 10^{-8}$). Physically nearest genes to lead GWAS variant per-peak are annotated per trait.

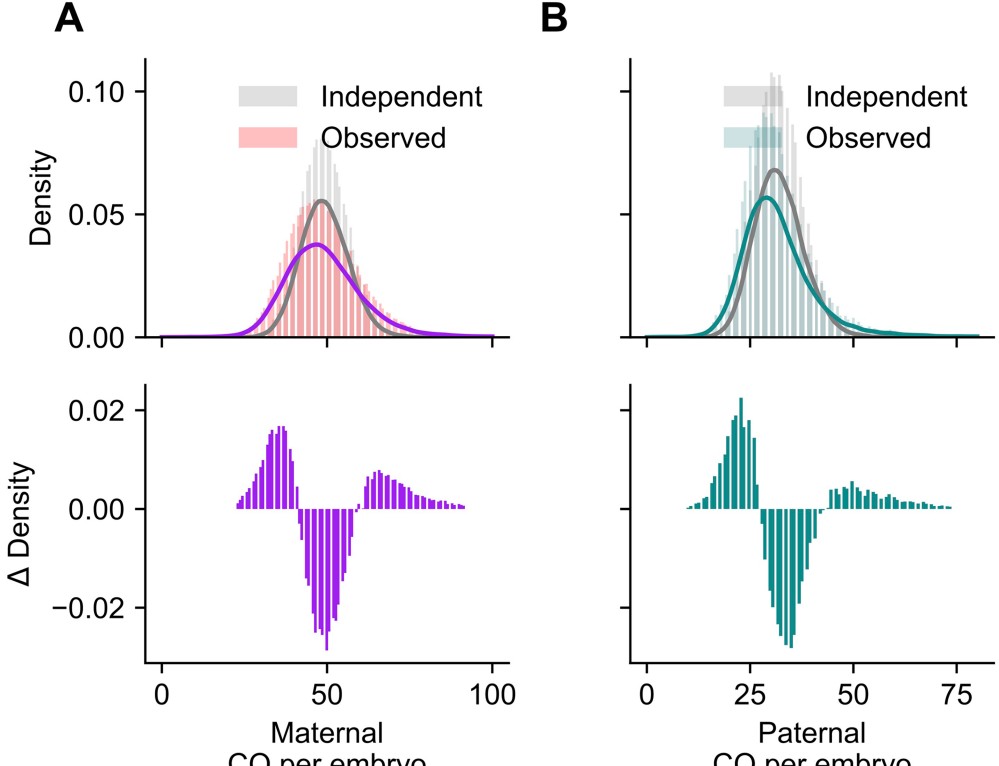

**Extended Data Fig. 7 | Cross-chromosomal covariance in crossover counts per embryo.** Comparison of observed crossover count (colored bars) against chromosomally independent simulations (gray bars) reflecting substantial overdispersion in crossovers due to positive covariance in (A) maternal and (B) paternal crossovers between chromosomes. Solid lines are kernel density estimates fit to the corresponding simulated and real crossover distributions.

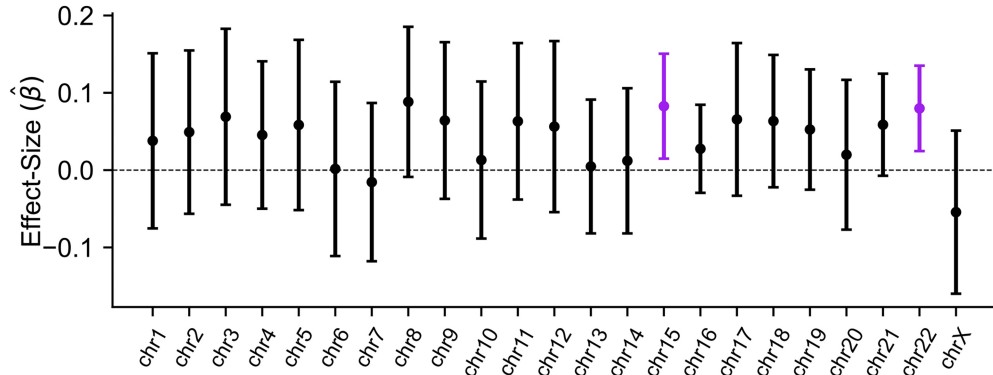

**Extended Data Fig. 8 | Marginal effect of rs6006737 on single-chromosome maternal meiotic aneuploidy.** For each test, embryos aneuploid for the chromosome of interest are compared to euploid embryos, excluding those with aneuploidies affecting other chromosomes. While the effect-size estimates are positive for nearly all chromosomes, only chromosomes 15 and 22 are individually significant.

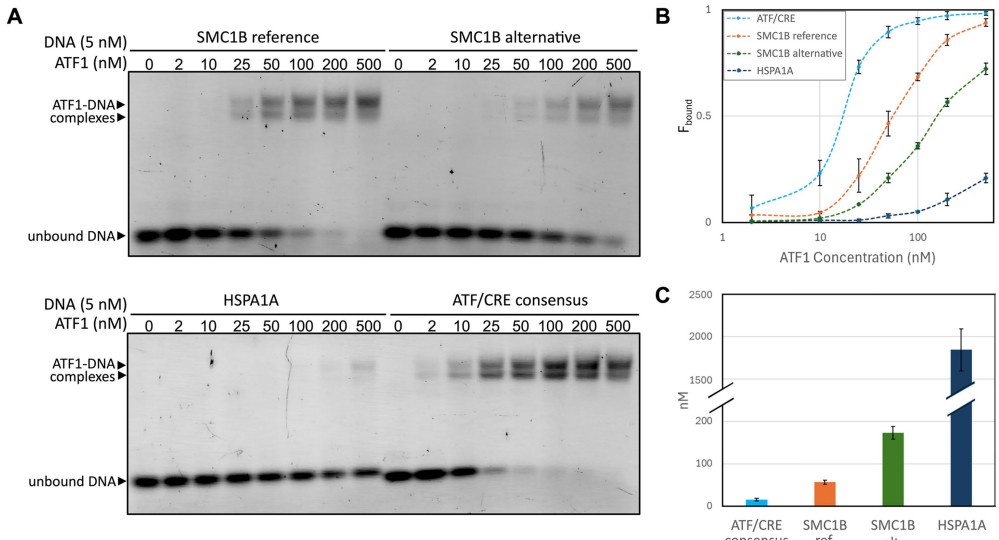

**Extended Data Fig. 9 | Electrophoretic mobility shift assay (EMSA) demonstrating the impact of SNP rs2272804 on binding of transcription factor ATF1 in vitro.** (A) Representative EMSA gel scans for *SMC1B* reference and alternative sequences, ATF/CRE consensus sequence from the Epstein-Barr virus *LMP1* gene promoter as a positive control, and *HSPA1A* promoter sequence as a negative control. ATF1 is a basic leucine zipper transcription factor; the two shifted ATF1-DNA complex bands indicate binding of ATF1 as a monomer or dimer. Raw gel scans across three replicates are shown in Fig. S13. (B) The fraction of DNA bound as ATF1 concentration was titrated. Error bars indicate standard deviation from 2–4 replicates for each concentration. (C) Values for the $K_D$ obtained from curve-fit (see Supplementary Methods). Error bars indicate standard deviation across replicates. Unbound DNA is of known sequence and length (*SMC1B* reference and alternative sequences, 30 bp; ATF/CRE positive control, 30 bp; *HSPA1A* negative control, 35 bp; see Supplementary Methods).

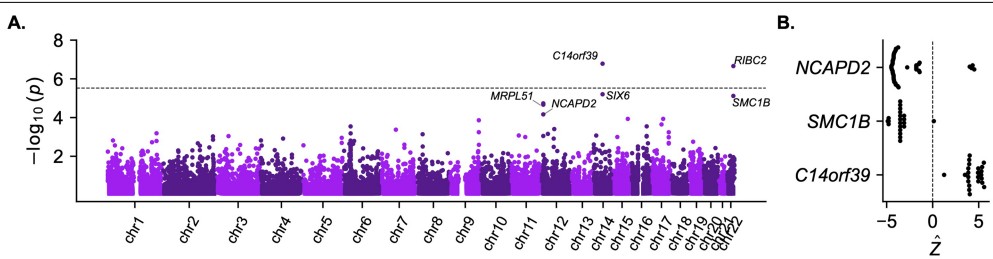

**Extended Data Fig. 10 | Transcriptome-wide association study (TWAS) for maternal meiotic aneuploidy.** (A) Transcriptome-wide association tests of maternal meiotic aneuploidy (two-sided binomial GLMM) and predicted maternal gene expression, combining across tissues (see Supplementary Methods). The dotted line indicates the threshold for transcriptome-wide significance ($p = 3 \times 10^{-6}$). (B) Per-tissue Z-scores indicating the direction of association between predicted expression and maternal meiotic aneuploidy.

# Reporting Summary

## Statistics

For all statistical analyses, confirm that the following items are present in the figure legend, table legend, main text, or Methods section.

| n/a | Confirmed | |
|---|---|---|
| ☐ | ☒ | The exact sample size (*n*) for each experimental group/condition, given as a discrete number and unit of measurement |
| ☐ | ☒ | A statement on whether measurements were taken from distinct samples or whether the same sample was measured repeatedly |
| ☐ | ☒ | The statistical test(s) used AND whether they are one- or two-sided *Only common tests should be described solely by name; describe more complex techniques in the Methods section.* |
| ☐ | ☒ | A description of all covariates tested |
| ☐ | ☒ | A description of any assumptions or corrections, such as tests of normality and adjustment for multiple comparisons |
| ☐ | ☒ | A full description of the statistical parameters including central tendency (e.g. means) or other basic estimates (e.g. regression coefficient) AND variation (e.g. standard deviation) or associated estimates of uncertainty (e.g. confidence intervals) |
| ☐ | ☒ | For null hypothesis testing, the test statistic (e.g. *F*, *t*, *r*) with confidence intervals, effect sizes, degrees of freedom and *P* value noted *Give P values as exact values whenever suitable.* |
| ☐ | ☒ | For Bayesian analysis, information on the choice of priors and Markov chain Monte Carlo settings |
| ☐ | ☒ | For hierarchical and complex designs, identification of the appropriate level for tests and full reporting of outcomes |
| ☐ | ☒ | Estimates of effect sizes (e.g. Cohen's *d*, Pearson's *r*), indicating how they were calculated |

*Our web collection on statistics for biologists contains articles on many of the points above.*

## Software and code

Policy information about availability of computer code

Data collection: No software was used for data collection, as data were received as raw X and Y genotype intensities for retrospective analysis.

Data analysis: Genotyping and imputation code is available on GitHub (https://github.com/mccoy-lab/natera_genotyping/) and archived on Zenodo (https://doi.org/10.5281/zenodo.17429676). Pipelines for inferring crossover recombination across sibling embryos is available on GitHub (https://github.com/mccoy-lab/natera_recomb) and archived on Zenodo (https://doi.org/10.5281/zenodo.17429678). Code for inferring aneuploidies and performing downstream analyses is available on GitHub (https://github.com/mccoy-lab/karyohmm; https://github.com/mccoy-lab/natera_aneuploidy) and archived on Zenodo (https://doi.org/10.5281/zenodo.17429669; https://doi.org/10.5281/zenodo.17429672).

Existing software packages used in the study include:
ACAT - version 0.91, https://github.com/yaowuliu/ACAT
BEAGLE - version 5.4, https://faculty.washington.edu/browning/beagle/b5_4.html
data.table - version 1.15.4, https://cran.r-project.org/package=data.table
Eagle - version 2.4.1, https://alkesgroup.broadinstitute.org/Eagle/
emmeans - version 1.10.3, https://rvlenth.github.io/emmeans/
LDlink - https://ldlink.nih.gov/
LD Score Regression, version 1.0.0 - https://github.com/bulik/ldsc
lme4 - version 1.1.35.5, https://cran.r-project.org/web/packages/lme4/index.html
MAST - version 4.11.4, meme-suite.org/
optiCall - version 0.8.1, www.well.ox.ac.uk/~gav/docs/optiCall/
PLINK - version 1.9, https://www.cog-genomics.org/plink/
R - version 4, https://www.r-project.org/

REGENIE - version 4.1, https://rgcgithub.github.io/regenie/
SINGER - version 0.1.8, https://github.com/popgenmethods/SINGER
tidyverse - version 2.0.0, https://tidyverse.org/

For manuscripts utilizing custom algorithms or software that are central to the research but not yet described in published literature, software must be made available to editors and reviewers. We strongly encourage code deposition in a community repository (e.g. GitHub). See the Nature Portfolio guidelines for submitting code & software for further information.

# Data

Policy information about availability of data

All manuscripts must include a data availability statement. This statement should provide the following information, where applicable:

- Accession codes, unique identifiers, or web links for publicly available datasets
- A description of any restrictions on data availability
- For clinical datasets or third party data, please ensure that the statement adheres to our policy

Association study summary statistics and aneuploidy and crossover calls are available on Zenodo: https://doi.org/10.5281/zenodo.15114528. Patient privacy and data use restrictions do not allow for external deposition of raw SNP microarray data. Researchers may request access to these data from Natera by contacting Zachary Demko (zdemko@natera.com). Initial responses to these requests can be expected within one month of receipt. Approved requests will require separate data use agreements in compliance with appropriate privacy laws.

Public data resources used in the study include:
GRCh37, GRCh38 (https://hgdownload.soe.ucsc.edu/)
GENCODE v37 (https://www.gencodegenes.org/human/release_37.html)
gnomAD (https://gnomad.broadinstitute.org/)
MAGE (https://github.com/mccoy-lab/MAGE/)
GTEx v8 (https://github.com/broadinstitute/gtex-v8)
ENCODE (https://www.encodeproject.org/)
deCODE genetic map (https://www.science.org/doi/10.1126/science.aau1043 - Supplementary File 1)
1000 Genomes Project + HGDP reference panel (https://github.com/atgu/hgdp_tgp)

# Research involving human participants, their data, or biological material

Policy information about studies with human participants or human data. See also policy information about sex, gender (identity/presentation), and sexual orientation and race, ethnicity and racism.

| Reporting on sex and gender | Mapping of meiotic crossovers and genome-wide association studies were performed by stratifying on the sex of the biological parents who provided eggs or sperm for in vitro fertilization, prior to preimplantation genetic testing. We use the term "biological parents" to refer to this set of individuals, the term "female" to refer to the biological sex of individuals who provided eggs, and the term "male" to refer to the biological sex of the individuals who provided sperm. While analyses were applied to both groups, a greater focus was placed on the female samples, as most chromosome abnormalities originate during female meiosis (egg formation), as confirmed in our study. |
|---|---|
| Reporting on race, ethnicity, or other socially relevant groupings | Geographically-defined population labels (European, African) are used in reference to allele frequencies of variants in external databases such as gnomAD and the 1000 Genomes Project, as defined in the original studies. To examine the ancestry distribution of the Natera sample, we performed principal component analysis (PCA) and interpreted results based on genetic similarity to 1000 Genomes samples with geographically-defined population labels. We report these results in the context of genetic similarity to reference samples, which is the accurate interpretation of PCA. |
| Population characteristics | The age distribution of biological parents is provided in the "Sample overview" section of the Methods. Detailed clinical histories of the biological parents are not available, but potential reasons for ordering the Natera Spectrum preimplantation genetic test, as listed on the sample requisition form, include: Balanced translocation or insertion in normal individual (Q95.0); Cystic Fibrosis gene carrier (Z14.1); Encounter for male factor infertility in female patient (Z31.81); Female infertility, unspecified (N97.9); Family history of carrier of genetic disease (Z84.81); Genetic carrier status, other than Cystic Fibrosis (Z14.8); Other screening for genetic and chromosomal anomalies (Z13.79); Recurrent pregnancy loss (N96); and Other. |
| Recruitment | This was a retrospective analysis of existing data, so there was no prospective recruitment. The data source is described in the "Data collection and sampling" section of the Methods. Briefly, this is a cohort of patients undergoing preimplantation genetic testing for a variety of referral reasons, including reasons related to fertility diagnoses (see above), so results must be interpreted in context of this ascertainment bias. |
| Ethics oversight | Research by Johns Hopkins was reviewed by the Johns Hopkins Homewood IRB (Not Human Subjects Research determination), while Natera received approval from Salus IRB (Category 4 Exempt Research). |

Note that full information on the approval of the study protocol must also be provided in the manuscript.

# Field-specific reporting

Please select the one below that is the best fit for your research. If you are not sure, read the appropriate sections before making your selection.

☒ Life sciences ☐ Behavioural & social sciences ☐ Ecological, evolutionary & environmental sciences

For a reference copy of the document with all sections, see nature.com/documents/nr-reporting-summary-flat.pdf

# Life sciences study design

All studies must disclose on these points even when the disclosure is negative.

| | |
|---|---|
| Sample size | All available samples (after quality control) were used in the study to maximize statistical power. |
| Data exclusions | Exclusion criteria for individual analyses are described in detail in the Methods. Small proportions of samples were excluded as technical outliers, as conclusions drawn from such data were deemed unreliable. For example, embryos with five or more nullisomic chromosomes were excluded, as these likely reflect DNA amplification failures. Chromosomes with ploidy calls supported by posterior probabilities below 0.9 were excluded to improve confidence in downstream results. Patients with fewer than three embryos were excluded, as chromosome-scale phasing could not be reliably determined in such cases. These exclusion criteria were not pre-specified prior to the study but were established based on theoretical considerations (e.g., phasing logic), simulation, and/or exploratory data analyses of technical variable distributions. All filtering was performed prior to biological hypothesis testing (e.g., GWAS analyses). |
| Replication | Samples for the genome-wide association study were randomly split into an independent discovery (85%) and test (15%) set for internal replication. |
| Randomization | All groups contrasted in our study were derived from the data itself, controlling for relevant covariates, as detailed in the Methods. For example, genome-wide association analyses compared genotypes among individuals with low versus high rates of embryonic aneuploidy (encoded as an overdispersed binomial quantitative trait), controlling for covariates such as maternal age, genotype principal components, etc. |
| Blinding | Blinding was not applicable because no experimental groups were assigned. Genotype-phenotype associations were analyzed retrospectively using existing data. |

# Reporting for specific materials, systems and methods

We require information from authors about some types of materials, experimental systems and methods used in many studies. Here, indicate whether each material, system or method listed is relevant to your study. If you are not sure if a list item applies to your research, read the appropriate section before selecting a response.

## Materials & experimental systems

| n/a | Involved in the study |
|---|---|
| ☒ ☐ | Antibodies |
| ☒ ☐ | Eukaryotic cell lines |
| ☒ ☐ | Palaeontology and archaeology |
| ☒ ☐ | Animals and other organisms |
| ☒ ☐ | Clinical data |
| ☒ ☐ | Dual use research of concern |
| ☒ ☐ | Plants |

## Methods

| n/a | Involved in the study |
|---|---|
| ☒ ☐ | ChIP-seq |
| ☒ ☐ | Flow cytometry |
| ☒ ☐ | MRI-based neuroimaging |

# Plants

| | |
|---|---|
| Seed stocks | *Report on the source of all seed stocks or other plant material used. If applicable, state the seed stock centre and catalogue number. If plant specimens were collected from the field, describe the collection location, date and sampling procedures.* |
| Novel plant genotypes | *Describe the methods by which all novel plant genotypes were produced. This includes those generated by transgenic approaches, gene editing, chemical/radiation-based mutagenesis and hybridization. For transgenic lines, describe the transformation method, the number of independent lines analyzed and the generation upon which experiments were performed. For gene-edited lines, describe the editor used, the endogenous sequence targeted for editing, the targeting guide RNA sequence (if applicable) and how the editor was applied.* |
| Authentication | *Describe any authentication procedures for each seed stock used or novel genotype generated. Describe any experiments used to assess the effect of a mutation and, where applicable, how potential secondary effects (e.g. second site T-DNA insertions, mosiacism, off-target gene editing) were examined.* |

