## [Peer Review File · Nature]

Common variation in meiosis genes shapes human recombination and aneuploidy

Corresponding Author: Dr Rajiv McCoy

Version 0:

Reviewer comments:

Referee #1

(Remarks to the Author)

The article studies the genetic factors shaping human recombination and aneuploidy risk. Whereas most previous studies have either used the healthy population to map human recombination or used fetal remains or live births to explore the causes of aneuploidy risks, this study is the first large scale analysis of aneuploid embryos before implantation, avoiding a large part of the selection and providing a less biased analysis. The analysis reveals SMC1B haplotypes are associated with crossover count and meiotic aneuploidy while TWAS identify candidate genes to regulate meiotic aneuploidy risk. The authors collect genomic information on an impressive number of embryos and their parents and this is the first study having direct access to such a large number of aneuploidies. The work is of importance to the field, strengthens existing hypotheses and provides some new genetic insights in the factors affecting female aneuploidy.

Comments

- Lower rates of recombination are observed, consistent with previous reports that advanced maternal ages correlate with reduced recombination rates. However, in younger women recombination rates have been shown to increase, especially in aneuploid embryos/trisomies. Are these women excluded, could this confound the results or, because of the nature of the embryos, is there a strong selection bias towards older women.
- The dataset is blastocysts. Since blastocyst underwent a first round of selection, the power might have been higher with cleavage stage embryo blastomeres. To what extent could the selection affect the results? Or looking from another angle: in the present study all aneuploidies are pooled, under the assumption that there are no chromosome specific genetic features that might affect aneuploidy rates. Was this investigated?
- Figure 1: aneuploidies primarily involve gain or loss of maternal homologs. However, the figure does not distinguish trisomies and monosomies. It might be valuable to know the ratio of monosomies/trisomies across the chromosomes. This could be included.
- Figure S1: age at time of collection across 22850 biological parents. In the method section it is stated that 'Maternal ages ranged from 20.1 to 55.8 years at the time of collection'. However, in the figure significant number of women have ages over 50 with outlier above 70. Hence, there seems to be a difference between the description in the methods and the figure. Could the authors comment/correct?
- The authors seem to assume that the four crossover-derived phenotypes (mean crossover count, hotspot occupancy, replication timing and GC content) are generally known. Except for mean crossover count, this may not be the case. Provide references.
- A link to the github (github.com/mccoy-lab/natera_genotyping) is provided for the genotype data. However, I could not access the site. Other github links are working. Is the genotype information freely accessible and/or how?

(Remarks on code availability)

A link to the github (github.com/mccoy-lab/natera_genotyping) is provided for the genotype data. Is the genotype information freely accessible and/or how? However, I could not access the site. Other github links are accessible. I did not test the code.

Referee #2

(Remarks to the Author)

This study analyzes aneuploidy and crossover patterns within a large dataset of embryo biopsies performed on IVF embryos

for the purpose of preimplantation genetic testing. Several loci associated with aneuploidy and crossover are identified—most of these replicate results that are known from the literature but one is a novel genome-wide association. Overall, the study is rigorously and carefully done, but I think it is also a bit conservative in staying away from commenting on the most interesting possible implications of the paper's results. Currently, I'd say the results in the paper have medium-high impact, but this impact could be increased if the authors went a bit more out on a limb to extrapolate what their results mean about the relationship between aneuploidy and embryonic health/fitness.

The fact that the paper identifies common alleles that are significantly associated with aneuploidy implies that alleles causing trophectoderm aneuploidy are not very damaging to fitness. This seems surprising if we assume that more aneuploidy means more pregnancy loss and that before modern miscarriage management, each pregnancy loss probably measurably increased the risk of maternal death. It seems like this paper contains enough information for the authors to come up with a lower bound on the historical selection coefficient of the SMC1B aneuploidy risk allele and then use this selection coefficient to bound the risk of pregnancy loss associated with each aneuploid embryo. For example, if every aneuploid embryo were inviable, would that be compatible with the finding of such a high frequency aneuploidy modifier? What about if each aneuploid embryo were only 50% as likely to be viable as a euploid embryo? Perhaps the lack of association between aneuploidy and clinical infertility is also definitive enough to put a bound on the strength of the relationship between embryo aneuploidy and viability. The relationship between aneuploidy detected in PGT and embryo viability is a topic of substantial interest and controversy, and many readers will likely weigh the impact of a paper like this by how successfully it moves the needle on our understanding of this issue.

I identified a few other places in the manuscript where some straightforward follow-up might additionally increase impact, as detailed below:

1. One striking feature of Figure 1C that is not mentioned in the text is that most paternal aneuploidies are sex chromosome aneuploidies and indeed that the majority of sex chromosome aneuploidies are paternal in origin. Has this been reported before? Do sex chromosome aneuploidies show the same qualitative associations with parental age and recombination rate as autosomal aneuploidies? It also seems ambiguous whether certain aneuploidies like an X0 karyotype should be represented as a chrX aneuploidy or a chrY aneuploidy, and this should be clarified.

2. Are the rates of specific aneuploidies (e.g. chr16 or chr21) repeated within families more often than expected under a model where aneuploidy rate varies among families but the affected chromosome is chosen at random from the population distribution? The recent Logson, et al. results on centromeres affecting Down syndrome rates suggests that some of the observed variation among families might be driven by cis-acting effects that elevate aneuploidy rates in a chromosome-specific way. The particular GWAS done in the paper would not be well powered to detect such effects, and a better test for cis-acting effects would be to say test all chr21 variants for association with elevated chr21 aneuploidy rates and do this for each chromosome. Given the sex chromosome results, it also seems warranted to test paternal sex chromosome variation for effects on paternal sex chromosome aneuploidy.

3. Is the number of embryos per family associated with any of the other metrics being tested in the study after controlling for parental age, for example, measures of infertility, the frequency of aneuploidy, or the recombination rate? Anecdotally, the number of embryos per IVF cycle seems to be regarded as a fertility metric, and if embryo aneuploidy causes a substantial rate of embryo loss we could expect to see a negative association between embryo count and aneuploidy rate.

4. The measures of infertility tested for associations in Figure 5 all appear to require clinical diagnosis. If possible, it might be more informative to test the traits in this study for association with a quantitative trait like number of children ever born, as has been done in several studies involving the UK Biobank and the CEPH pedigrees (notably Cawthon, et al. Scientific Reports 2020).

5. Can monogenic or polygenic selection tests be used to support the claim that recombination rate is likely under stabilizing selection? Would selection against a high recombination rate ever be invoked before physical crossover interference imposed an upper limit on the recombination rate?

Additional minor comments:

1. The maternal age curve in Figure 1D looks marginally more cubic than quadratic given how the data points curve downward at the top. Did this authors test this and find that a quadratic regression was actually preferable?

2. Does this dataset replicate previous findings of a positive association between female recombination rate and female fertility? (Coop, et al. Science 2008)

3. The data availability statement should give more information on what is required to access the data—will it be available on dbGaP since the study is NIH funded?

Referee #3

(Remarks to the Author)

The work by Carioscia et al. cleverly utilizes genotyping data from preimplantation genetic testing from fertilized embryos to examine the rate of aneuploidy and its genetic basis. Previous studies were limited by only studying surviving individuals

with aneuploidies. Importantly, the authors also identified crossovers from the same samples, enabling a direct evaluation of the potential influence of recombination on aneuploidy. They confirmed the association with maternal age and determined that aneuploid embryos have a significantly lower number of crossovers. They then investigated the genetic component associated with aneuploidy risk and found an association with a haplotype including the meiotic cohesin component, SMC1B. The authors further explore available datasets to determine that the aneuploidy risk allele is associated with lower expression of SMC1B. This association is further refined to identify two specific variants located within non-coding regions. The study identifies shared genetic variants influencing both the rate of female meiotic recombination and aneuploidy, suggesting a common genetic origin for these factors and their impact on pregnancy loss.

This elegant study represents a significant advancement in understanding the genetic factors influencing recombination rate variability and its link to aneuploidy risk. The paper is well-written, the presentation is excellent, and the conclusions are thoroughly supported by the data. I believe it will be of great interest to the meiosis and reproductive biology communities.

Comments:

Fig 1D: Wouldn't a U-shaped curve be expected? As shown by: Gruhn et al., Science (2019)?

Can E0 (non recombinant chromosomes) be detected in this system? Or would they be confused with the non-recombinant sister that segregated?

Figure 3B and line 235. SMC1B is a meiotic specific gene and shouldn't be expressed in LCLs. I appreciate that there is a low level of expression (at least from the GTEx browser) but it would be useful to comment on this. Regarding the TWAS analysis of recombination phenotypes and maternal aneuploidies using all GTEx tissues, have the authors considered restricting the expression data to testes? Although it doesn't represent fetal ovary expression, testes data is relevant to meiotic gene expression.

It is surprising to me that a variant with a deleterious effect on chromosome segregation (and fertility) is present at such high frequency in the population. Is there any explanation?

A variant in SYCE2 was recently shown to be associated with associates with pregnancy loss and recombination (<https://doi.org/10.1038/s41594-023-01209-y>) Have you looked at this variant in your dataset?

Is the dataset representative of the general population, or is there a discernible bias stemming from the initial requirement of seeking IVF treatment?

I cannot comment on the appropriateness of the statistical testing in each scenario, but it would be useful for the reader if the rationale behind choosing models is explained. Is it dependent on the expected distribution of the underlying data? Or you tried many and these are the best fit?

From the methods "After calling genotypes, we lifted over variants from human genome build GRCh37 to GRCh38, which resulted in the removal of 1,831 variants. Following application of these filters, we retained 275,425 variants across the genome". Why are the variants detected using GRCh37?

(Remarks on code availability)

The Github repository is well-organized with clear readme files. The scripts examined were well-commented and structured. This review was cursory and did not involve installation or execution of the code.

Version 1:

Reviewer comments:

Referee #1

(Remarks to the Author)

Thank you for the extensive answers to the questions. All questions have been properly addressed.

Referee #2

(Remarks to the Author)

The authors have done a thorough job with revisions and the manuscript is looking very strong. I just have a minor additional comment concerning the new theoretical exploration of the fitness consequences of the aneuploidy-increasing variant. The authors are assuming that the only fitness consequence of the variant is a reduction in the production of viable embryos, resulting in a ~1% lifetime fertility reduction. However, even in modern times, first trimester pregnancy loss sometimes results in medical complications requiring hospitalization (a recent study found at <https://pmc.ncbi.nlm.nih.gov/articles/PMC6507132/> estimates about 50 early-miscarriage-associated hospitalizations per 1000 deliveries). Earlier in human history, these complications of miscarriage may have been fatal or resulted in long term loss of fertility. I suggest that the authors incorporate this maternal morbidity cost into their theoretical model and see whether it then becomes necessary to incorporate stronger fitness effect attenuation to explain the neutral segregation of the

aneuploidy-associated variant.

(Remarks on code availability)

I did not review the code in detail or attempt to install and run it, but I looked at each of the above repositories. They each have a README file that appears to be clear and detailed, making the paper's core analyses likely to be reproducible.

Referee #3

(Remarks to the Author)

The authors have done a commendable job addressing my and the other reviewers' comments. The addition of an in-depth exploration of the evolutionary context of the risk variant is a bonus that strengthens an already excellent work. This manuscript is a valuable addition to the literature and will be important in the field.

(Remarks on code availability)

The Github repository is well-organized with clear readme files. The scripts examined were well-commented and structured. This review was cursory and did not involve installation or execution of the code.

Version 2:

Reviewer comments:

Referee #2

(Remarks to the Author)

I appreciate the authors' thoughtful response and have no remaining criticisms of the manuscript.

Response to Reviewers for Carioscia, Biddanda, et al. 2025 submission to *Nature*

Reviewer comments in black

Author responses in blue

Referee #1

The article studies the genetic factors shaping human recombination and aneuploidy risk. Whereas most previous studies have either used the healthy population to map human recombination or used fetal remains or live births to explore the causes of aneuploidy risks, this study is the first large scale analysis of aneuploid embryos before implantation, avoiding a large part of the selection and providing a less biased analysis. The analysis reveals SMC1B haplotypes are associated with crossover count and meiotic aneuploidy while TWAS identify candidate genes to regulate meiotic aneuploidy risk. The authors collect genomic information on an impressive number of embryos and their parents and this is the first study having direct access to such a large number of aneuploidies. The work is of importance to the field, strengthens existing hypotheses and provides some new genetic insights in the factors affecting female aneuploidy.

Thank you for this thorough and encouraging synopsis of our work, as well as the constructive comments that follow.

Comments

1. Lower rates of recombination are observed, consistent with previous reports that advanced maternal ages correlate with reduced recombination rates. However, in younger women recombination rates have been shown to increase, especially in aneuploid embryos/trisomies. Are these women excluded, could this confound the results or, because of the nature of the embryos, is there a strong selection bias towards older women.

This comment raises several interesting but nuanced points that we hope to clarify here. One key observation that we report in our study is that the number of crossovers is lower in aneuploid versus euploid embryos, even when restricting to the normal disomic chromosomes. This pattern is independent of maternal age (maternal age is included as a covariate; see next paragraph) and is consistent with an average protective role of crossovers on aneuploidy risk. Notably, crossovers on trisomic chromosomes are undetectable in PGT data when both reciprocal chromatids produced by a single crossover event are transmitted to the embryo. By relying on the positive covariance in crossover count across chromosomes (Wang et al. 2019) (a pattern that we replicate), our statistical approach overcomes this limitation by focusing on

disomic chromosomes of aneuploid embryos, providing more robust evidence of the relationship between crossovers and aneuploidy.

The second point regards the relationship between the number of crossovers and maternal age, which we now directly address in the text (lines 158-168), copied below:

Previous literature has provided conflicting evidence about the relationship between counts of meiotic crossovers and maternal age, with some studies reporting a positive association^{22,23,39} and others reporting a negative association^{40,41}. As those studies largely focused on living families, positive associations were typically interpreted as evidence of selection against aneuploid embryos, which possess fewer crossovers on average and increase in frequency with maternal age. Within our sample, we observed no significant association between maternal age and number of maternal crossovers (Poisson GLMM, $\beta = -2.62 \times 10^{-5}$, $SE=1.68 \times 10^{-3}$, $p = 0.988$). Importantly, this observation held even when restricting analysis to euploid embryos (Poisson GLMM, $\beta = 5.12 \times 10^{-4}$, $SE=1.43 \times 10^{-3}$, $p = 0.721$), offering a point of evidence against the hypothesis that embryonic aneuploidy is responsible for a positive association between crossovers and maternal age.

Returning to the reviewer's question about the maternal age distribution of our study cohort, we refer to **Figure S1**. The age range of patients is 20.1 to 55.8 years, with the bulk of the distribution >35 years, as expected for this patient population. Nevertheless, given the overall scale of the data, there are hundreds of individuals in the lower and upper parts of the maternal age range, providing statistical power for testing hypotheses about age relationships (e.g., see significant maternal age effects on even rare autosomal aneuploidies). All of our key analyses control for maternal age, and in the case of the *SMC1B* association, we also tested for potential interactions between genotype and maternal age, ensuring the robustness of our conclusions.

1. The dataset is blastocysts. Since blastocyst underwent a first round of selection, the power might have been higher with cleavage stage embryo blastomeres. To what extent could the selection affect the results? Or looking from another angle: in the present study all aneuploidies are pooled, under the assumption that there are no chromosome specific genetic features that might affect aneuploidy rates. Was this investigated?

This is a very interesting point, and indeed our previous work, as well as other research in the field, has documented strong selection prior to blastocyst formation against complex forms of aneuploidy that originate from abnormal (e.g., multipolar) cell divisions that are common in human IVF embryos (McCoy, Demko, Ryan, Banjevic, Hill, Sigurjonsson, Rabinowitz, and Petrov 2015; McCoy et al. 2023). Our previous research used a smaller PGT dataset from cleavage-stage embryos to discover genetic variants that influence these complex mitotic aneuploidies (McCoy, Demko, Ryan, Banjevic, Hill, Sigurjonsson, Rabinowitz, Fraser, et al. 2015). The clinical field of PGT has since transitioned toward blastocyst biopsy as a safer and more effective approach (Scott et al. 2013). Our study is thus well positioned to identify potential associations between maternal genotype and maternal meiotic aneuploidy risk, without the

confounding effects of high-level mitotic aneuploidies that are more common in cleavage-stage embryo biopsies (Rana et al. 2023).

The point about definition of the aneuploidy phenotype (individual chromosomes versus combined) is interesting and is something we considered. We had previously concluded that chromosome-stratified analyses would be underpowered, but the reviewer's idea to restrict the genotype data to the corresponding chromosome (i.e., *cis* effects) offers an intriguing approach to limit the multiple testing burden. We therefore followed this suggestion, and while we identified no genome-wide significant associations in these chromosome-stratified *cis*-association tests, we now briefly note this result in the text (lines 226-229) and provide full summary statistics in our online data repository.

2. Figure 1: aneuploidies primarily involve gain or loss of maternal homologs. However, the figure does not distinguish trisomies and monosomies. It might be valuable to know the ratio of monosomies/trisomies across the chromosomes. This could be included.

Thank you for this suggestion. We have updated **Figure 1C** to separately depict counts of trisomies and monosomies affecting each chromosome.

3. Figure S1: age at time of collection across 22850 biological parents. In the method section it is stated that 'Maternal ages ranged from 20.1 to 55.8 years at the time of collection'. However, in the figure significant number of women have ages over 50 with outlier above 70. Hence, there seems to be a difference between the description in the methods and the figure. Could the authors comment/correct?

Thank you for catching this inconsistency, which involves the distinction between "biological parents" and patients who used egg or sperm donors. We have now revised **Figure S1** to only include patient-partner pairs that did not use egg or sperm donors while clarifying this point in the in-text reference and **Figure S1** caption.

4. The authors seem to assume that the four crossover-derived phenotypes (mean crossover count, hotspot occupancy, replication timing and GC content) are generally known. Except for mean crossover count, this may not be the case. Provide references.

This is a good suggestion for improving readability of the text. These phenotypes were previously defined in several studies by the research group at deCODE, and our definition of hotspot occupancy is borrowed from Coop et al., 2008. We now briefly define these phenotypes at first use within the main text (lines 170-175), while referring to Methods section "Genome-wide and transcriptome-wide association with recombination phenotypes" where more detailed descriptions are provided.

5. A link to the github (github.com/mccoy-lab/natera_genotyping) is provided for the genotype data. However, I could not access the site. Other github links are working. Is the genotype information freely accessible and/or how?

We apologize that this repository was inaccessible, and we have now made it public. While the GitHub repository does not contain raw genotype data (see below), it provides the comprehensive genotyping pipeline that we used to process the raw SNP array intensity data supplied by Natera.

Remarks on code availability

A link to the github (github.com/mccoy-lab/natera_genotyping) is provided for the genotype data. Is the genotype information freely accessible and/or how? However, I could not access the site. Other github links are accessible. I did not test the code.

As discussed with the editor, patient privacy and data use restrictions preclude us from depositing raw genotype data externally, even in a controlled-access repository. To maximize transparency and reproducibility, we have publicly released individual embryo-level processed results data (aneuploidy and crossover calls), complete GWAS summary statistics, as well as documented analysis code. The latter includes our newly developed software for inferring aneuploidy and recombination events from pedigree-based data (karyoHMM), which researchers can apply to related datasets. Finally, we provide a point of contact at Natera for researchers who seek to access the raw data through Natera. This would require a separate data use agreement, which would need to comply with appropriate privacy laws. These data restrictions are similar to related studies in the *Nature* family of journals (e.g., (Arnadottir et al. 2025; Janssen et al. 2024)), so we have modeled our revised Data Availability statement on these examples.

Referee #2

This study analyzes aneuploidy and crossover patterns within a large dataset of embryo biopsies performed on IVF embryos for the purpose of preimplantation genetic testing. Several loci associated with aneuploidy and crossover are identified—most of these replicate results that are known from the literature but one is a novel genome-wide association. Overall, the study is rigorously and carefully done, but I think it is also a bit conservative in staying away from commenting on the most interesting possible implications of the paper's results. Currently, I'd say the results in the paper have medium-high impact, but this impact could be increased if the authors went a bit more out on a limb to extrapolate what their results mean about the relationship between aneuploidy and embryonic health/fitness.

The fact that the paper identifies common alleles that are significantly associated with aneuploidy implies that alleles causing trophoctoderm aneuploidy are not very damaging to fitness. This seems surprising if we assume that more aneuploidy means more pregnancy loss and that before modern miscarriage management, each pregnancy loss probably measurably increased the risk of maternal death. It seems like this paper contains enough information for

the authors to come up with a lower bound on the historical selection coefficient of the *SMC1B* aneuploidy risk allele and then use this selection coefficient to bound the risk of pregnancy loss associated with each aneuploid embryo. For example, if every aneuploid embryo were inviable, would that be compatible with the finding of such a high frequency aneuploidy modifier? What about if each aneuploid embryo were only 50% as likely to be viable as a euploid embryo? Perhaps the lack of association between aneuploidy and clinical infertility is also definitive enough to put a bound on the strength of the relationship between embryo aneuploidy and viability. The relationship between aneuploidy detected in PGT and embryo viability is a topic of substantial interest and controversy, and many readers will likely weigh the impact of a paper like this by how successfully it moves the needle on our understanding of this issue.

Thank you very much for the encouraging comments about the rigor and impact of our work, as well as highlighting the opportunity to expand upon the evolutionary implications. In addition to the novelty of the *SMC1B* association, we wish to emphasize the downstream functional genomic dissection of this locus, as well as associations with additional components of meiotic machinery revealed by TWAS. In addition, while our analyses relating crossover count to aneuploidy risk support an existing hypothesis within the field, our novel statistical approach (GLMM contrasting crossover counts on disomic chromosomes of euploid vs. aneuploid embryos) overcomes a persistent technical limitation that impacted most previous studies.

We would like to clarify one point regarding the relationship between aneuploidy and embryo viability. In contrast to some of our previous work on mosaic aneuploidies (which are more common in cleavage-stage embryo biopsies), the aneuploidies detected by SNP array genotyping of blastocyst-stage embryo biopsies are of almost exclusive maternal meiotic origin and are therefore not restricted to the trophectoderm but would affect the entire embryo (Rana et al. 2023). This is supported by the observations of 1) the extreme bias toward extra/missing maternal versus paternal homologs (**Fig. 1C**), 2) the diagnostic meiotic trisomy signature of "both parental homologs" from the mother (**Fig. S20**), and 3) the strong maternal age association with maternal aneuploidy (**Fig. 1D**). Nearly all of these maternal meiotic-origin aneuploidies, which affect the entire embryo, will result in pregnancy loss (Tiegs et al. 2021). Most of these pregnancy losses occur very early in gestation (e.g., implantation failure), but meiotic aneuploidies are also the leading cause of clinical pregnancy loss. A small minority of meiotic aneuploidies are survivable (e.g., Trisomy 21) but are associated with developmental syndromes.

Nevertheless, the suggestion to provide more evolutionary context is well taken. We agree that a common variant with seemingly harmful impacts on reproduction poses an interesting evolutionary puzzle that we can address more directly. To this end, we have added a Results section "*Evolutionary modeling of aneuploidy risk variants at SMC1B*" that extends our empirical results and proposes a theoretical model to explain the origins and maintenance of a high frequency aneuploidy-associated allele.

Briefly, our results suggest that (excluding alternative models such as balancing selection, which we also discuss) the relationship between the total number of viable embryos that a woman

could potentially produce over her reproductive lifespan and realized fitness must have been sufficiently weak over the course of human history to allow the risk allele to drift to high frequency (i.e., effective neutrality; $s < 1 / 2N_e$). We provide theoretical bounds on this relationship while also examining sensitivity to the assumed reproductive timespan. As one empirical point of support, which we added to the PheWAS section, we note that the aneuploidy-associated variant of *SMC1B* exhibits no significant association with number of children ever born in data from the UK Biobank, although we recognize that this does not formally rule out the existence of such an association in human evolutionary history. Overall, we believe that the neutral explanation is most parsimonious.

I identified a few other places in the manuscript where some straightforward follow-up might additionally increase impact, as detailed below:

Comments

1. One striking feature of Figure 1C that is not mentioned in the text is that most paternal aneuploidies are sex chromosome aneuploidies and indeed that the majority of sex chromosome aneuploidies are paternal in origin. Has this been reported before? Do sex chromosome aneuploidies show the same qualitative associations with parental age and recombination rate as autosomal aneuploidies? It also seems ambiguous whether certain aneuploidies like an X0 karyotype should be represented as a chrX aneuploidy or a chrY aneuploidy, and this should be clarified.

Thank you for these suggestions. You are correct that most sex chromosome aneuploidies are of paternal origin and that most paternal aneuploidies affect the sex chromosomes. This has been reported in previous studies of human sperm, including based on single-cell DNA sequencing (Bell et al. 2020) **TEXT REDACTED**

FIGURE REDACTED

We now specifically note this pattern in the text and reference the above study. Interestingly, however, the paternal aneuploidies exhibit no associations with paternal age, even when restricting to paternal aneuploidies of the sex chromosomes ($\beta = 0.002$, $p = 0.914$).

You are also correct that certain aneuploidies such as X0 are ambiguous with regard to X vs. Y. To address this point, we have updated **Figure 1C** to simply list all of the various abnormal sex

chromosome complements as a stacked bar labeled as a single chromosome category ("Sex chroms.").

2. Are the rates of specific aneuploidies (e.g. chr16 or chr21) repeated within families more often than expected under a model where aneuploidy rate varies among families but the affected chromosome is chosen at random from the population distribution? The recent Logson, et al. results on centromeres affecting Down syndrome rates suggests that some of the observed variation among families might be driven by cis-acting effects that elevate aneuploidy rates in a chromosome-specific way. The particular GWAS done in the paper would not be well powered to detect such effects, and a better test for cis-acting effects would be to say test all chr21 variants for association with elevated chr21 aneuploidy rates and do this for each chromosome. Given the sex chromosome results, it also seems warranted to test paternal sex chromosome variation for effects on paternal sex chromosome aneuploidy.

This is a very interesting point, also raised by Reviewer 1.

The question about variance within/between families can be addressed by extending the overdispersion test that we had previously applied to pooled aneuploidies to aneuploidies of individual chromosomes. As with the pooled phenotype, we observe that even after controlling for maternal age, aneuploidies of these individual chromosomes are significantly overdispersed compared to binomial expectations, supporting genetic and/or environmental factors modulating their risk. We now report these chromosome-stratified aneuploidy dispersion parameters in a **Table S4**.

These results motivate the *cis*-genetic association tests suggested by the reviewer, whereby we test for associations between variants on a given chromosome and aneuploidies of that chromosome. While none of the results from these scans achieved genome-wide significance, we now briefly note this result in the text (lines 226-229) and provide full summary statistics in our online data repository.

3. Is the number of embryos per family associated with any of the other metrics being tested in the study after controlling for parental age, for example, measures of infertility, the frequency of aneuploidy, or the recombination rate? Anecdotally, the number of embryos per IVF cycle seems to be regarded as a fertility metric, and if embryo aneuploidy causes a substantial rate of embryo loss we could expect to see a negative association between embryo count and aneuploidy rate.

This is a very interesting point. In practice, we have to be cautious about interpreting the phenotype of "embryo count", which can be strongly influenced by clinical decisions that are unknown to us. For example, we would hypothesize that patients who produce a high proportion of aneuploid embryos in a given cycle will be more likely to pursue an additional cycle. This is substantiated in our results by the observation that when combining across cycles and controlling for relevant covariates including maternal age, there is a significant *positive association* between aneuploidy rate and embryo count ($\beta = 0.045$, $p = 2.17 \times 10^{-16}$). However,

when fitting a mixed effect linear model that does not combine cycles, but includes a random effect of patient to account for covariance across cycles, there is a significant *negative association* between aneuploidy rate and embryo count ($\beta = -0.030$, $p = 1.26 \times 10^{-5}$). We now report the latter finding in the text, as we believe that this model is more appropriate for testing the relevant biological hypothesis raised by the reviewer. This is also consistent with our previous work showing that while complex mitotic aneuploidies are the main cause of embryo arrest prior to the blastocyst stage, there is also evidence of selection against aneuploidies of maternal meiotic origin during the cleavage stages (McCoy et al. 2023).

The relationship between recombination rate and embryo count is much more sensitive to technical confounding. Specifically, crossovers are identified and validated by contrasting patterns of transmission across sibling embryos. We restrict our crossover analyses to families with ≥ 3 sibling embryos. Even after imposing this restriction, we anticipate that our specificity for detecting crossovers increases with an increasing number of sibling embryos, as supported by our simulation studies. As such, although we observe a negative association between number of called crossovers and number of embryos within the data, we cannot ascribe this to a biological cause. Given this potential source of confounding, all of our crossover-related analyses (e.g., contrasting between aneuploid and euploid embryos) always include the number of embryos as a numerical covariate.

4. The measures of infertility tested for associations in Figure 5 all appear to require clinical diagnosis. If possible, it might be more informative to test the traits in this study for association with a quantitative trait like number of children ever born, as has been done in several studies involving the UK Biobank and the CEPH pedigrees (notably Cawthon, et al. Scientific Reports 2020).

Thank you for this suggestion. We have updated our PheWAS results (**Figure 5**) to include published GWAS of (1) number of children ever born and (2) childlessness from (Mathieson et al. 2023). We observe no significant associations between the aneuploidy and recombination-associated variants and either of these traits, which is also interesting in the context of the evolutionary theory we have added upon revision.

5. Can monogenic or polygenic selection tests be used to support the claim that recombination rate is likely under stabilizing selection? Would selection against a high recombination rate ever be invoked before physical crossover interference imposed an upper limit on the recombination rate?

This is a very interesting question regarding the evolution of recombination, which we have explored internally during revision. In response, we estimated the overall SNP-heritability of sex-specific recombination traits using GCTA and explored the extent of variation in recombination *within* versus *between* individuals, for individuals with a large number of euploid embryos.

One signature of both negative directional selection and stabilizing selection is an enrichment of rare variant heritability (Schoech et al. 2019; Koch and Sunyaev 2021). However, when estimating heritability stratified by allele frequency within our dataset using the GREML-LDMS framework (Yang et al. 2015), the resulting estimates have wide standard errors which limit any evaluation of enrichment or depletion, even for recombination traits with overall non-zero estimates of SNP-heritability (see figures below). We attribute this to 1) poorer imputation accuracy for rare variants and 2) lack of sufficient sample size for estimating stratified heritability for a trait with modest overall heritability. Additionally, the phenotype measured for “Recombination Rate” is the *mean recombination rate* across *all* of the individuals embryos, and using individuals with > 10 euploid embryos, we can see that the variation of crossover counts *within* an individual is substantially greater than *between* individuals, consistent with low narrow-sense heritability (despite the GCTA estimates).

Considering these results together, we believe that fully addressing the question of stabilizing selection on recombination would require assessment in cohorts with parental whole-genome sequencing data and development of new empirical tests or theoretical models. While beyond the scope of the current aneuploidy-focused study, we hope to revisit this topic in future research.

Nevertheless, we have expanded our discussion of potential constraints on recombination rates at either end of the distribution. One hypothesis is that the double-stranded breaks (DSBs) initiating crossover recombination are mutagenic, inducing single-nucleotide substitutions as well as small indels and structural variants (Halldorsson et al. 2019; Hinch, Donnelly, and Hinch 2023). Some such de novo mutations are expected to be deleterious (Racimo and Schraiber 2014), which may impose an *upper limit* on double-stranded breaks (and consequently crossovers). The influence of physical crossover interference, likely also contributes to the *upper limit* of resolved crossovers after DSBs are placed. The following statement in our discussion section speaks to these hypothesized effects (lines 502 - 512):

The observation that alleles associated with lower rates of recombination are associated with higher rates of aneuploidy raises interesting questions about the evolutionary forces that shape recombination and aneuploidy within and between species. In addition to generating new combinations of alleles, research has demonstrated that recombination is mutagenic, inducing point mutations and structural variation near hotspots of double-strand breaks^{22,86}. These observations together suggest a model of stabilizing selection, whereby rates of recombination may be constrained on the lower and upper ends to limit aneuploidy and other classes of deleterious mutations, respectively. More comprehensive models of recombination rate evolution must also consider mechanical constraints such as crossover interference⁸⁷, which reduces occurrence of nearby crossovers, as well as the role of crossovers in facilitating adaptation⁸⁸.

Minor comments

1. The maternal age curve in Figure 1D looks marginally more cubic than quadratic given how the data points curve downward at the top. Did this authors test this and find that a quadratic regression was actually preferable?

The sigmoid shape of the curve comes from the logit link function used in the binomial GLMM, so it would have a similar visual form on this scale even with only a linear term for maternal age (on the logit scale). Nevertheless, the reviewer raises a valid question about model selection. Given the scale of the data, sequential F-tests on nested models are statistically significant up to a fifth degree polynomial term, but we believe that such a model lacks biological justification and likely overfits the data. To address this concern, we therefore performed 10-fold cross-validation, evaluating average binomial deviance on the held-out folds for increasingly complex models ranging from polynomial degrees of 0 (i.e., no maternal age effect) to 6 (i.e., a sixth degree polynomial term). We observed that beyond the quadratic term, the change in binomial deviance was negligible (see figure below), and we opted to retain the quadratic model according the "one standard error rule", which selects the most parsimonious model within one standard error of the minimum cross-validation error (Hastie, Tibshirani, and Friedman 2009). We now include a methods section where we describe this model selection procedure and reference the below supplementary figure (**Fig S5**).

2. Does this dataset replicate previous findings of a positive association between female recombination rate and female fertility? (Coop, et al. Science 2008)

We find that individual-specific rates of recombination were inversely associated with aneuploidy (after controlling for maternal age and our discovered genetic associations), suggesting that this previously-reported effect is likely at least partially driven by the reduction in aneuploidy risk.

We initially hypothesized that we could use the number of euploid embryos observed per mother as a proxy for female fertility, but the relationship between recombination rate and embryo count is sensitive to technical confounding. Specifically, crossovers are identified and validated by contrasting patterns of transmission across sibling embryos. We restrict our crossover analyses to families with ≥ 3 sibling embryos. Even after imposing this restriction, we anticipate that our specificity for detecting crossovers increases with an increasing number of sibling embryos, as supported by our simulation studies.

As such, although we observe a negative association between number of called crossovers and number of embryos within the data, we cannot ascribe this to a biological cause. Given this potential source of confounding, all of our crossover-related analyses (e.g., contrasting between aneuploid and euploid embryos) always include the number of embryos as a numerical covariate, and we instead focus on aneuploidy (see first paragraph above) as the more reliable fertility trait to associate with recombination rate.

3. The data availability statement should give more information on what is required to access the data—will it be available on dbGaP since the study is NIH funded?

Patient privacy and data use restrictions preclude us from depositing raw genotype data externally, even in a controlled-access repository. As the data were generated for clinical

purposes by Natera and were not prospectively generated with NIH funding, the NIH Genomic Data Sharing Policy is not applicable.

Nevertheless, we are committed to maximizing transparency, rigor, and reproducibility of our work within the above legal and privacy constraints. To this end, we have publicly released individual embryo-level processed results data (aneuploidy and crossover calls), complete GWAS summary statistics, as well as documented analysis code. The latter includes our newly developed software for inferring aneuploidy and recombination events from pedigree-based data (karyoHMM), which researchers can apply to related datasets. Finally, we provide a point of contact at Natera for researchers who seek to access the raw data through Natera; this would require a separate data use agreement, which would need to comply with appropriate privacy laws. These data restrictions are similar to related studies in the *Nature* family of journals (e.g., (Arnadottir et al. 2025; Janssen et al. 2024)), so we have modeled our revised Data Availability statement on these examples.

Referee #3

The work by Carioscia et al. cleverly utilizes genotyping data from preimplantation genetic testing from fertilized embryos to examine the rate of aneuploidy and its genetic basis. Previous studies were limited by only studying surviving individuals with aneuploidies. Importantly, the authors also identified crossovers from the same samples, enabling a direct evaluation of the potential influence of recombination on aneuploidy. They confirmed the association with maternal age and determined that aneuploid embryos have a significantly lower number of crossovers. They then investigated the genetic component associated with aneuploidy risk and found an association with a haplotype including the meiotic cohesin component, SMC1B. The authors further explore available datasets to determine that the aneuploidy risk allele is associated with lower expression of SMC1B. This association is further refined to identify two specific variants located within non-coding regions. The study identifies shared genetic variants influencing both the rate of female meiotic recombination and aneuploidy, suggesting a common genetic origin for these factors and their impact on pregnancy loss.

This elegant study represents a significant advancement in understanding the genetic factors influencing recombination rate variability and its link to aneuploidy risk. The paper is well-written, the presentation is excellent, and the conclusions are thoroughly supported by the data. I believe it will be of great interest to the meiosis and reproductive biology communities.

We are grateful for this thorough and encouraging review of our work, as well as the thoughtful questions and suggestions that follow.

Comments

1. Fig 1D: Wouldn't a U-shaped curve be expected? As shown by: Gruhn et al., Science (2019)?

The data reported in Gruhn et al. (2019) extend to younger maternal ages than are present in our dataset. Specifically, their study included women < 20 years old, whereas the minimum maternal age within our current sample is 22.1 years (**Fig. S1**), limiting our statistical power to formally test this effect.

However, the question about how we selected an appropriate statistical model is still valid and was also raised by Reviewer 2. In short, we performed 10-fold cross-validation, evaluating average binomial deviance on the held-out folds for models ranging from polynomial degrees of 0 (i.e., no maternal age effect) to 6 (i.e., a sixth degree polynomial term). We observed that beyond the quadratic term, the change in binomial deviance was negligible, and we therefore elected to retain the quadratic model according to the "one-standard-error-rule" (Hastie, Tibshirani, and Friedman 2009) to balance the bias-variance tradeoff and enhance interpretability (**Fig. S5**).

The sigmoid shape of the quadratic curve depicted in **Figure 1D** comes from the logit link function used in the binomial GLMM, so it would have a similar visual form on this scale even with only a linear term for maternal age (on the logit scale). Meanwhile, this quadratic model is indeed U-shaped (curving slightly upward below 24 years), but **Figure 1D** restricts the range ages with at least 25 individuals, and we are wary of overinterpreting these extremes or relaxing these thresholds given our limited sample size.

2. Can E0 (non recombinant chromosomes) be detected in this system? Or would they be confused with the non-recombinant sister that segregated?

This is an excellent question. While our GLMM supports the conclusion that aneuploid embryos have fewer crossovers on average, we did not directly investigate E0 chromosomes. We now tabulate raw counts of chromosomes without a detected crossover within inferred euploid embryos, briefly noting our results in the text and referencing a supplemental figure that we copy below (**Fig. S6**). We observe qualitatively similar trends with the previous literature, where shorter chromosomes have higher proportions of no detected crossovers (Fledel-Alon et al. 2009).

While these measurements are likely correlated with true E0 rates per-chromosome, crossovers at chromosome ends (e.g., telomeric/subtelomeric regions) can be missed due to the low density of the SNP array. Additionally, for shorter chromosomes which may only experience one or two crossovers per homolog, these observations are consistent with a high probability of observing a non-recombinant non-sister chromatid (even though these chromosomes have undergone an exchange). For our figure, we therefore refer to our observations as “chromosomes with no detected crossover”. However, with revised methodological development, we believe that inferring E0 chromosomes in our dataset is possible (building on (Fiedel-Alon et al. 2009)), and we plan to pursue this goal in future work.

3. Figure 3B and line 235. *SMC1B* is a meiotic specific gene and shouldn't be expressed in LCLs. I appreciate that there is a low level of expression (at least from the GTEx browser) but it would be useful to comment on this. Regarding the TWAS analysis of recombination phenotypes and maternal aneuploidies using all GTEx tissues, have the authors considered restricting the expression data to testes? Although it doesn't represent fetal ovary expression, testes data is relevant to meiotic gene expression.

Thank you for this careful observation. We agree that expression of *SMC1B* is low in LCLs compared to testes or what might be expected for fetal ovaries (although also see (Mannini et al. 2015)).

The TWAS results that we report in the paper are based on the combined p-values across all tissues within GTEx, including testes, using the aggregated Cauchy association test (ACAT), which allows for arbitrary correlation structure. However, we have also provided the single-tissue TWAS results in **Table S3**, which show that the *SMC1B* signal we detect is consistent in, but not exclusive to, the testes. We note that tissues in which a prediction model could not be obtained from PrediXcan (e.g., because a gene is not sufficiently expressed) are excluded (and denoted as NA in the above table). As *cis*-genetic effects on expression tend to be correlated across

tissues (e.g., by impacting the promoter), we believe that combining results across all tissues provides the greatest power to discover associations.

The focus on LCLs for fine-mapping extends from the observation that the *SMC1B* signal is consistent across tissues, but takes advantage of the population genetic diversity of the MAGE dataset, which exclusively consists of LCLs (n = 731). The greater diversity of this sample, which spans 26 globally distributed populations from the 1000 Genomes Project, captures more recombination events in its history; this has the effect of breaking up linkage disequilibrium and increasing resolution for fine-mapping, beyond what is possible within the predominantly European GTEx cohort. The high resolution for fine-mapping allowed us to isolate a single putative causal SNP defining an *SMC1B* credible set, in turn motivating our downstream experiments to test its impacts on transcription factor binding *in vitro*.

4. It is surprising to me that a variant with a deleterious effect on chromosome segregation (and fertility) is present at such high frequency in the population. Is there any explanation?

This is a very interesting question that was also raised by Reviewer 1. Our most extensive revisions to the manuscript have involved adding a Results section on evolutionary modeling ("Evolutionary modeling of aneuploidy risk variants at *SMC1B*"), where we explore the parameter space that would enable this variant to achieve high frequency. We believe these results provide valuable evolutionary context that can broadly inform understanding of genetic variation affecting human fertility.

5. A variant in *SYCE2* was recently shown to be associated with associates with pregnancy loss and recombination (<https://doi.org/10.1038/s41594-023-01209-y>) Have you looked at this variant in your dataset?

We are aware of this reported association. Because the variant is rare in the broader European-ancestry population beyond Iceland (rs189296436; gnomAD EUR AF 0.5%), we could not impute the variant with high enough confidence (dosage $R^2 > 0.8$) to satisfy our filtering criteria for use in GWAS. We anticipate that future analyses using exome sequencing may be better suited to examine the effects of this and other rare variants on aneuploidy risk.

6. Is the dataset representative of the general population, or is there a discernible bias stemming from the initial requirement of seeking IVF treatment?

Unfortunately, because Natera is a reference lab as opposed to a fertility clinic, the clinical metadata for addressing this question is very incomplete and unstandardized. Broadly, we can say that patients in this dataset pursued either (or both) of PGT-A (preimplantation genetic testing for aneuploidy) or PGT-M (preimplantation genetic testing for monogenic disorders). Patients seeking PGT-M are generally carriers of monogenic disease mutations but do not have known fertility challenges. Possible (non-mutually exclusive) reasons for PGT-A or PGT-M referral include:

- Monogenic disease (e.g., cystic fibrosis) genetic carrier status

- Balanced translocation or insertion carrier status
- Male infertility
- Female infertility
- Advanced maternal age
- Previous IVF failure
- Recurrent pregnancy loss

Past work on this topic showed that while certain referral categories are statistically enriched for aneuploidy, the effect sizes are modest (McCoy, Demko, Ryan, Banjevic, Hill, Sigurjonsson, Rabinowitz, and Petrov 2015).

Related to the point above, we also confirmed that the aneuploidy and recombination-associated variants identified in our study (e.g., rs6006737) exhibit similar frequencies within our dataset when compared to external ancestry-matched panels such as the European population of the 1000 Genomes Project.

7. I cannot comment on the appropriateness of the statistical testing in each scenario, but it would be useful for the reader if the rationale behind choosing models is explained. Is it dependent on the expected distribution of the underlying data? Or you tried many and these are the best fit?

Thank you for this suggestion. We have added more detail regarding model selection for the association between incidence of maternal meiotic aneuploidy and maternal age (lines 1133-1146, **Fig. S5**). Throughout the Methods, we have also added additional justification for various models (e.g., Poisson models for count-based response variables and binomial models for proportion based response variables).

8. From the methods “After calling genotypes, we lifted over variants from human genome build GRCh37 to GRCh38, which resulted in the removal of 1,831 variants. Following application of these filters, we retained 275,425 variants across the genome”. Why are the variants detected using GRCh37?

The original HumanCytoSNP array was designed based on GRCh36 (hg18). However, the array manifests provided to us by Natera were already lifted to GRCh37 (hg19) coordinates. Processing the CytoSNP data on GRCh37 thus avoided an additional liftover step and ensured consistency with the provided resources. While we could have alternatively lifted over these resource files to GRCh38, this would have limited practical effect on variant inclusion or analysis outcomes, as the 1,831 affected variants would simply be excluded earlier in the pipeline.

Remarks on code availability

The Github repository is well-organized with clear readme files. The scripts examined were well-commented and structured. This review was cursory and did not involve installation or execution of the code.

Thank you for looking at the repository!

References

- Arnadóttir, Gudny A., Hakon Jonsson, Tanja Schlaikjær Hartwig, Jennifer R. Gruhn, Peter Loof Møller, Arnaldur Gylfason, David Westergaard, et al. 2025. "Sequence Diversity Lost in Early Pregnancy." *Nature* 642 (8068): 672–81.
- Bell, Avery Davis, Curtis J. Mello, James Nemesh, Sara A. Brumbaugh, Alec Wysoker, and Steven A. McCarroll. 2020. "Insights into Variation in Meiosis from 31,228 Human Sperm Genomes." *Nature* 583 (7815): 259–64.
- Fledel-Alon, Adi, Daniel J. Wilson, Karl Broman, Xiaoquan Wen, Carole Ober, Graham Coop, and Molly Przeworski. 2009. "Broad-Scale Recombination Patterns Underlying Proper Disjunction in Humans." *PLoS Genetics* 5 (9): e1000658.
- Halldorsson, Bjarni V., Gunnar Palsson, Olafur A. Stefansson, Hakon Jonsson, Marteinn T. Hardarson, Hannes P. Eggertsson, Bjarni Gunnarsson, et al. 2019. "Characterizing Mutagenic Effects of Recombination through a Sequence-Level Genetic Map." *Science* 363 (6425). <https://doi.org/10.1126/science.aau1043>.
- Hastie, T., R. Tibshirani, and J. Friedman. 2009. *The Elements of Statistical Learning: Data Mining, Inference, and Prediction, Second Edition*. 2nd ed. Springer Series in Statistics. New York, NY: Springer.
- Hinch, Robert, Peter Donnelly, and Anjali Gupta Hinch. 2023. "Meiotic DNA Breaks Drive Multifaceted Mutagenesis in the Human Germ Line." *Science* 382 (6674): eadh2531.
- Janssen, Anouk E. J., Rebekka M. Koeck, Rick Essers, Ping Cao, Wanwisa van Dijk, Marion Drüsedau, Jeroen Meekels, et al. 2024. "Clinical-Grade Whole Genome Sequencing-Based Haplarithmisis Enables All Forms of Preimplantation Genetic Testing." *Nature Communications* 15 (1): 7164.
- Koch, Evan M., and Shamil R. Sunyaev. 2021. "Maintenance of Complex Trait Variation: Classic Theory and Modern Data." *Frontiers in Genetics* 12 (November):763363.
- Mannini, Linda, Francesco Cucco, Valentina Quarantotti, Clelia Amato, Mara Tinti, Luigi Tana, Annalisa Frattini, et al. 2015. "SMC1B Is Present in Mammalian Somatic Cells and Interacts with Mitotic Cohesin Proteins." *Scientific Reports* 5 (1): 18472.
- Mathieson, Iain, Felix R. Day, Nicola Barban, Felix C. Tropf, David M. Brazel, eQTLGen Consortium, BIOS Consortium, et al. 2023. "Genome-Wide Analysis Identifies Genetic Effects on Reproductive Success and Ongoing Natural Selection at the FADS Locus." *Nature Human Behaviour* 7 (5): 790–801.
- McCoy, Rajiv C., Zachary P. Demko, Allison Ryan, Milena Banjevic, Matthew Hill, Styrmir Sigurjonsson, Matthew Rabinowitz, and Dmitri A. Petrov. 2015. "Evidence of Selection against Complex Mitotic-Origin Aneuploidy during Preimplantation Development." *PLoS Genetics* 11 (10): e1005601.
- McCoy, Rajiv C., Zachary Demko, Allison Ryan, Milena Banjevic, Matthew Hill, Styrmir

- Sigurjonsson, Matthew Rabinowitz, Hunter B. Fraser, and Dmitri A. Petrov. 2015. "Common Variants Spanning *PLK4* Are Associated with Mitotic-Origin Aneuploidy in Human Embryos." *Science* 348 (6231): 235–38.
- McCoy, Rajiv C., Michael C. Summers, Abeo McCollin, Christian S. Ottolini, Kamal Ahuja, and Alan H. Handyside. 2023. "Meiotic and Mitotic Aneuploidies Drive Arrest of in Vitro Fertilized Human Preimplantation Embryos." *Genome Medicine* 15 (1): 77.
- Racimo, Fernando, and Joshua G. Schraiber. 2014. "Approximation to the Distribution of Fitness Effects across Functional Categories in Human Segregating Polymorphisms." *PLoS Genetics* 10 (11): e1004697.
- Rana, Bhavini, Kevin Lambrese, Robert Mendola, Jia Xu, John Garrisi, Kathleen Miller, Diego Marin, and Nathan R. Treff. 2023. "Identifying Parental and Cell-Division Origins of Aneuploidy in the Human Blastocyst." *The American Journal of Human Genetics* 110 (4): 565–74.
- Schoech, Armin P., Daniel M. Jordan, Po-Ru Loh, Steven Gazal, Luke J. O'Connor, Daniel J. Balick, Pier F. Palamara, Hilary K. Finucane, Shamil R. Sunyaev, and Alkes L. Price. 2019. "Quantification of Frequency-Dependent Genetic Architectures in 25 UK Biobank Traits Reveals Action of Negative Selection." *Nature Communications* 10 (1): 790.
- Scott, Richard T., Jr, Kathleen M. Upham, Eric J. Forman, Tian Zhao, and Nathan R. Treff. 2013. "Cleavage-Stage Biopsy Significantly Impairs Human Embryonic Implantation Potential While Blastocyst Biopsy Does Not: A Randomized and Paired Clinical Trial." *Fertility and Sterility* 100 (3): 624–30.
- Tiegs, Ashley W., Xin Tao, Yiping Zhan, Christine Whitehead, Julia Kim, Brent Hanson, Emily Osman, et al. 2021. "A Multicenter, Prospective, Blinded, Nonselection Study Evaluating the Predictive Value of an Aneuploid Diagnosis Using a Targeted next-Generation Sequencing-Based Preimplantation Genetic Testing for Aneuploidy Assay and Impact of Biopsy." *Fertility and Sterility* 115 (3): 627–37.
- Wang, Shunxin, Carl Veller, Fei Sun, Aurora Ruiz-Herrera, Yongliang Shang, Hongbin Liu, Denise Zickler, Zijiang Chen, Nancy Kleckner, and Liangran Zhang. 2019. "Per-Nucleus Crossover Covariation and Implications for Evolution." *Cell* 177 (2): 326–38.e16.
- Yang, Jian, Andrew Bakshi, Zhihong Zhu, Gibran Hemani, Anna A. E. Vinkhuyzen, Sang Hong Lee, Matthew R. Robinson, et al. 2015. "Genetic Variance Estimation with Imputed Variants Finds Negligible Missing Heritability for Human Height and Body Mass Index." *Nature Genetics* 47 (10): 1114–20.

Response to Reviewers for Carioscia, Biddanda, et al. 2025 submission to *Nature* (second revision)

Reviewer comments in black

Author responses in blue

Referee #1

Remarks to the Author

Thank you for the extensive answers to the questions. All questions have been properly addressed.

Thank you for the thorough and constructive comments on our work.

Referee #2

Remarks to the Author

The authors have done a thorough job with revisions and the manuscript is looking very strong. I just have a minor additional comment concerning the new theoretical exploration of the fitness consequences of the aneuploidy-increasing variant. The authors are assuming that the only fitness consequence of the variant is a reduction in the production of viable embryos, resulting in a ~1% lifetime fertility reduction. However, even in modern times, first trimester pregnancy loss sometimes results in medical complications requiring hospitalization (a recent study found at <https://pmc.ncbi.nlm.nih.gov/articles/PMC6507132/> estimates about 50 early-miscarriage-associated hospitalizations per 1000 deliveries). Earlier in human history, these complications of miscarriage may have been fatal or resulted in long term loss of fertility. I suggest that the authors incorporate this maternal morbidity cost into their theoretical model and see whether it then becomes necessary to incorporate stronger fitness effect attenuation to explain the neutral segregation of the aneuploidy-associated variant.

The reviewer raises an important point that the fitness impacts of aneuploidy may arise from mechanisms beyond embryo survival, including via effects of miscarriage on maternal survival and future fertility.

In an initial response to this comment, we drafted a model incorporating probabilities of implantation, miscarriage, and complications of miscarriage, conditional on euploid vs. aneuploid conception. Although literature estimates of several of these parameters are

unavailable or highly uncertain (especially over the course of human evolution), our preliminary analyses indicated that these modeled secondary effects on fitness are small compared to the direct impact on embryo viability. However, upon further consideration and discussion with colleagues, we recognized that this is just one specific pathway by which pregnancy outcomes can affect fitness. Pregnancy, childbirth, and miscarriage may influence fitness in complex ways, including via maternal survival, future fertility, and parental or grandparental care—subjects of extensive research in the field of life history theory. To accurately reflect this complexity, we elected to retain the original simpler model and address these additional pathways to fitness effects through description and citation of relevant research in the field (lines 445-456):

However, while the number of euploid embryos a woman can produce is presumably correlated with fitness, it may constitute only a weak proxy, as realized fitness is also determined by myriad stochastic, environmental, and behavioral factors largely independent of genotype. Moreover, pregnancy, childbirth, and miscarriage can influence fitness in complex ways, including via impacts on maternal survival, future fertility, and parental/grandparental care⁵². We observed that to reach the theoretical threshold for evading negative selection, the realized selection coefficient (s) must be scaled by a factor (α) ≤ 0.01 , relative to s_{proxy} (see Supplementary Note). While the historical relationship between s and s_{proxy} is unknown, their weak correlation in contemporary populations is evidenced by the lack of association between the aneuploidy risk allele and fertility phenotypes such as number of children ever born and childlessness⁵³ (Fig. 5). These results highlight the inadequacy of simplistic proxies of fitness—a limitation long appreciated in the field of life history theory⁵⁴.

In summary, we are grateful to the reviewer for drawing our attention to the complex pathways through which aneuploidy can impact fitness, and while we have not presented a detailed mathematical extension (as this risks over-emphasizing one specific pathway), we believe our revisions address the spirit of the comment by offering additional context and reference to the broader life history literature.

Remarks on code availability

I did not review the code in detail or attempt to install and run it, but I looked at each of the above repositories. They each have a README file that appears to be clear and detailed, making the paper's core analyses likely to be reproducible.

Thank you for looking over our code repository.

Referee #3

Remarks to the Author

The authors have done a commendable job addressing my and the other reviewers' comments. The addition of an in-depth exploration of the evolutionary context of the risk variant is a bonus that strengthens an already excellent work. This manuscript is a valuable addition to the literature and will be important in the field.

Thank you for the thorough and constructive comments on our work.

Referee #3 (Remarks on code availability):

The Github repository is well-organized with clear readme files. The scripts examined were well-commented and structured. This review was cursory and did not involve installation or execution of the code.

Thank you also for looking over our code repository.